# A canonical trajectory of executive function maturation from adolescence to adulthood

Brenden Tervo-Clemmens ®[1,2,3] ✉, Finnegan J. Calabro[4,5], Ashley C. Parr ®[4], Jennifer Fedor ®[4,6], William Foran ®[4] & Beatriz Luna[3,4,5]

Theories of human neurobehavioral development suggest executive functions mature from childhood through adolescence, underlying adolescent risk-taking and the emergence of psychopathology. Investigations with relatively small datasets or narrow subsets of measures have identified general executive function development, but the specific maturational timing and independence of potential executive function subcomponents remain unknown. Integrating four independent datasets (N = 10,766; 8–35 years old) with twenty-three measures from seventeen tasks, we provide a precise charting, multi-assessment investigation, and replication of executive function development from adolescence to adulthood. Across assessments and datasets, executive functions follow a canonical non-linear trajectory, with rapid and statistically significant development in late childhood to mid-adolescence (10–15 years old), before stabilizing to adult-levels in late adolescence (18–20 years old). Age effects are well captured by domain-general processes that generate reproducible developmental templates across assessments and datasets. Results provide a canonical trajectory of executive function maturation that demarcates the boundaries of adolescence and can be integrated into future studies.

Adolescence is a unique period of the lifespan, initiated by puberty and characterized by the maturation of cognitive, affective, and social processes that culminate in a transition to independence and adulthood[1–3]. Among maturational processes, theories from neuroscience and psychology have placed a particular emphasis on the development of goal-directed cognitive abilities (e.g., response inhibition, working memory, task-switching, and planning behaviors) that are hypothesized to index a common process of executive function or cognitive control[4–6]. In parallel to socioemotional development and environmental influences, a protracted maturation[7,8] and/or stabilization of executive function[9] into adulthood has been suggested to contribute to lifespan peaks in risk-taking behaviors (e.g., substance use initiation[10]; though see also refs. 11,12) and increased vulnerability to psychiatric disorders[13] during adolescence. Ongoing executive function changes during adolescence have been used in colloquial, legal[9,14], and scientific (see[4] for review) contexts to differentiate adolescents from adults and clarify adolescence as a period of continued development.

Adolescent executive function development has been studied in relatively small (N's ~200[15,16]) independent investigations using a broad range of tasks or in relatively large studies (N's ~ 1000[17,18]) using very narrow assessments of executive function. No large-scale, multi-assessment, multi-dataset reproducibility investigations of adolescent executive function development have been performed. Further, common analytic approaches do not quantitatively define maturational timing and/or plateaus toward adult-levels of performance. The magnitude of executive function changes during adolescence, the precise timing of when adolescents reach adult-levels, and the

[1]Department of Psychiatry & Behavioral Sciences, University of Minnesota, Minneapolis, MN, USA. [2]Masonic Institute for the Developing Brain, University of Minnesota, Minneapolis, MN, USA. [3]Department of Psychology, University of Pittsburgh, Pittsburgh, PA, USA. [4]Department of Psychiatry, University of Pittsburgh, Pittsburgh, PA, USA. [5]Department of Bioengineering, University of Pittsburgh, Pittsburgh, PA, USA. [6]Department of Biostatistics, University of Pittsburgh, Pittsburgh, PA, USA. ✉e-mail: btervocl@umn.edu

potential diversity of processes assessed by varying executive function tasks, thus remains widely debated.

Empirical research suggests that while adolescents can perform complex, goal-directed behaviors that rely on executive functions, their performance is not as accurate or as fast as adults[5,15,16,19–24]. Age-related increases in correct response rates (accuracy) and decreases in the speed of responses (i.e., latency/reaction time) have been observed for a broad range of laboratory-based and neuropsychological executive function tasks (e.g., working memory, response inhibition, switching, planning) during adolescence (see refs. [1,25] for reviews). Theoretical models built from these observations and related observations in animal studies[26], as well as broader[27] and historical perspectives of psychological development[28], have led to hypotheses suggesting that cognitive development continues through adolescence and may reach maturity in the second decade of life (e.g., by 20 years old[9,29]) or later (e.g., ≥ 25 years old[8,27]) in humans. A range of methodological, analytic, and data availability challenges, however, have thus far prevented direct and comprehensive testing of the maturational timing of adolescent executive function development and the specific age when executive functions reach adult-levels. Nevertheless, understanding not just whether behaviors are changing with age, but also their shape and form, is fundamental to developmental science[15,30–32] and corresponding health policies and intervention/prevention strategies for adolescents. Defining the shape and form of cognitive development likewise has key implications for research on mechanisms of ongoing (potentially critical period) plasticity and factors influencing the opening and closing of the adolescent period[3].

There are unique challenges to defining the normative maturational timing of adolescent executive function development that arises from multiple sources, including inter-individual differences among participants and across datasets, difficulties in designing analytic frameworks that directly assess maturational timing[33], and potential variability among the many tasks designed to assess executive function[5,34]. The first of these challenges is beginning to be addressed through larger study designs (e.g., Nathan Kline Institute-Rockland Sample[35] [NKI], National Consortium on Alcohol and Neurodevelopment in Adolescence[36] [NCANDA], Philadelphia Neurodevelopmental Cohort[37] [PNC]) and data aggregation techniques, as increasing dataset sizes and the inclusion of multiple datasets can better overcome sampling variability[38,39] to estimate generalizable normative developmental trajectories. Addressing the latter challenges, however, requires conceptual and methodological advancements.

Initial investigations in adolescent research have often relied on a fixed-developmental shape (e.g., linear, inverse linear, quadratic regression models) or categorical comparisons (e.g., adolescents versus children/adults) to identify age-related differences[32,33]. While essential to establish that age-related differences in executive function generally occur during adolescence, such fixed-developmental, parametric comparisons prevent the systematic investigation of the relative rate and timing of maturation that is essential for precise developmental science. Such approaches likewise have prevented resolution of foundational theories of adolescent neurobehavioral development, where distinct linear and non-linear shapes have been proposed[4]. Therefore, while prominent theories suggest adolescents may reach adult-levels of executive function between 20- and 25 years old, such plateaus in developmental change have not been investigated in most empirical research and are not testable within commonly used analytic frameworks. This lack of resolution on the maturational timing of adolescent executive function also poses challenges for related lifespan research, where a potentially distinct developmental concept of emerging adulthood (~18–25 years old[40]) has likewise been justified, in part, by potential ongoing cognitive changes. New methods (e.g., general additive models[41]) that can quantitatively define the potentially non-linear developmental

trajectory of executive function during adolescence, as well as multiple large publicly available datasets, now allow for precise estimation of the maturational timing of executive function.

A further challenge to defining the maturational timing of executive function development arises from the potential variability among the many tasks designed to assess executive function. Empirical studies have often focused on an individual, or a relatively narrow subset, of tasks (see for example discussion in refs. [5,42]). Fewer studies have therefore investigated the developmental similarity among potential subprocesses indexed by the dozens of laboratory-based and neuropsychological executive function measures used in the broader literature. While there is a growing use of standardized tasks (e.g., Delis-Kaplan Executive Function System[43] [D-KEFS], Cambridge Neuropsychological Test Automated Battery[44] [CANTAB], Penn Computerized Neurocognitive Battery[45] [Penn CNB]), systematic comparisons across these instruments are similarly limited. Many neurodevelopmental and psychological theories[4,7–9] emphasize a broad unitary process of executive function development, suggesting the maturation of performance on any one of these tasks may generalize to broader executive function development. However, alternative perspectives have also been proposed. Prior work in adults (both healthy college students[42], as well as patients with frontal lobe damage[46]), for example, has suggested that executive function tasks support a unity/diversity framework, where commonality and correlation are observed amongst all executive function measures (unity), but individual aspects of executive function maintain a degree of separability (diversity). Owing to the focus on individual functions and tasks or narrow subsets in most adolescent research, it nevertheless remains unclear whether adolescent executive function development is driven by multiple, independent processes (diversity) and/or the maturation of a more common domain-general process (unity).

Here we aggregate four large-scale, independent datasets to construct a comprehensive set of executive function data spanning the entire adolescent period as well as the relative transitional periods of late childhood and early adulthood (total age range: 8–35, total $N = 10,766$, total visits = 13,817) that includes 23 executive function measures from 17 distinct tasks. In addition to large-scale replication, we directly address prior challenges in defining the maturational timing and domain-generality of adolescent executive function development with multiple large independent cohorts (two longitudinal, two cross-sectional), non-linear modeling approaches that directly define maturational timing, and the inclusion of a broad executive function battery that permit the investigation of both potential unitary and diversity processes. Taken together, this work identifies a canonical non-linear developmental trajectory of executive function maturation that generalizes across datasets and assessments, with rapid age-related change from late childhood to early adolescence (10–15 years old), small but significant changes in mid-adolescence (15–18 years old), before stabilizing to adult-levels in late adolescence (18–20 years old). The similarity in developmental trajectories is well accounted for by domain-general processes consistent with theories of unitary executive function and fluid cognition. The insights and data developed here can inform neuroscientific and psychological theories of the adolescent period and guide future translational research in health and disease.

## Results

### Executive function development follows a canonical trajectory across datasets and tasks

Participants ranging from 8–35 years old (Supplementary Fig. S1) were drawn from two large longitudinal studies of executive function development, including data collected by our group (Luna, $N = 196$, total visits = 666) and data collected as part of the National Consortium on Alcohol and Neurodevelopment (NCANDA[36], $N = 831$, total visits = 3412), as well as two large cross-sectional studies, including

data collected as part of the Nathan Kline Institute-Rockland Sample (NKI[35], $N = 588$), and data from the Philadelphia Neurodevelopmental Cohort (PNC[37], $N = 9151$). Studies relied on community-based samples from across the United States (see Methods) that were balanced for biological sex at birth and in the aggregate, were consistent with national patterns of race and ethnicity (Supplementary Table S1). Family income varied both within and between datasets, but as in previous reports across behavioral sciences[47], was generally higher than national averages (Supplementary Table S1). Secondary analyses however, suggested the sample composition of included datasets well approximated broader population patterns for primary results (See Supplementary Methods, Supplementary Fig. S2). Across the studies, participants performed a variety of executive function tasks (twenty-three measures from seventeen distinct EF tasks; Supplementary Table S2), including those designed to measure processes of response inhibition (e.g., Antisaccade, Stroop), working memory (e.g., Spatial Span), planning (e.g., Stockings of Cambridge), as well as those from standard computerized neurocognitive batteries (Penn Computerized Neurocognitive Battery[45]) that include tasks designed to measure executive function (e.g., Conditional Exclusion Test, N-Back Test, Continuous Performance Test[34]) and a neuropsychological executive function battery (Delis-Kaplan Executive Function System [D-KEFs]: Tower, Trails, Design Fluency, Color-Word Interference). Response types included button presses, eye movements, and experimenter-administered neuropsychological performance (e.g., D-KEFs). For most tasks, both latency (speed of responses) and accuracy (e.g., correct response rate) measures were available (see also Methods).

We first examined the developmental trajectory of each executive function measure independently using non-linear regression models with penalized splines (general additive mixed models (GAMM) for longitudinal data; general additive models (GAM) for cross-sectional data: see Methods). Unlike the fixed-developmental shape approaches that are typically used in adolescent research, this allowed us to estimate flexible, data-driven trajectories and explore the shape of development (functional form of age) for each executive function measure. These analyses revealed that nearly all executive function measures (20/23 measures) had corrected significant (corrected $p$'s < 0.004, [two-sided], calculated via default procedures of GAM that performs an equality test of parameters of the smoothed term to zero[48]; see Supplementary Table S3 for full statistics as well reproducible variable names from public datasets) age-related differences and followed a similar non-linear developmental trajectory, with rapid development in late childhood to mid-adolescence (10–15 years old), smaller changes through mid-adolescence (15–18 years old), before stabilizing to adult-levels in late adolescence (18–20 years old) (Fig. 1A–D). Critically, age-related differences were observed across nearly all tasks from all four independent datasets, with accuracy measures showing significant age-related increases and latency measures showing parallel significant age-related decreases (Fig. 1A–D). The average total age-related change (max-min of GAM/GAMM fits) was large based on conventional effect size standards (mean across all measures from all datasets in standard deviation [z] units: 1.38; Fig. 1A–D). Overlapping visualization of all measures with significant age-related differences from all datasets further highlights a potential canonical shape of normative adolescent executive function development (Fig. 1E).

### Executive function significantly develops through late adolescence
To precisely quantify periods of significant developmental differences and estimate when measures reached adult-levels, we next examined the local slope (first derivative) of age-related differences across all ages in 1/10th of year intervals for all non-linear GAMM/GAM models. As in prior developmental research in other domains[49,50], a simulation approach (10,000 iterations) was used to construct confidence intervals for the first derivative of the fitted models to assess statistically significant age-related differences at each age ($p < 0.05$ [two-sided] via simultaneous confidence intervals[51] to account for multiple tests across ages: see Methods). Age-ranges in which the simultaneous 95% confidence interval of the first derivative of the GAM/GAMM fits did not include zero ($p < 0.05$, two-sided) were classified as statistically significant. We note that a thresholded 95% confidence interval (an unthreholded version can be viewed in full in Supplementary Fig. S3), instead of for example exact p-values, are provided here as in previous work to highlight age ranges of statistical significance[49,50]. Consistent with theoretical models of adolescence, significant ($p < 0.05$ [two-sided] via simultaneous confidence intervals) age-related changes in executive function accuracy (increases) and latency (decreases) were observed during early to middle adolescent periods (10–15 years old) for nearly all measures (Fig. 2A–D). Effect size benchmarks do not yet exist for short-timescale developmental changes, however given the short span of age examined via the derivative (units scaled to per-year change) and the total age-related changes (Fig. 1A–E), local effect sizes are judged to be large (e.g., mean z unit change from 10–15 years old: .142 per-year [accuracy]; −.175 per-year [latency], Fig. 2E; see Fig. 2A–D for z unit scaling for all measures). From middle to late adolescent periods (15–18 years old), smaller but still statistically significant ($p < 0.05$ [two-sided] via simultaneous confidence intervals) changes were observed for several measures (Fig. 2A–D). After late adolescence (>18 years old), very few measures exhibited statistically significant ($p < 0.05$ [two-sided] via simultaneous confidence intervals) age-related change (Fig. 2A–D).

Aggregate analysis across measures and tasks (three-level point-wise meta-analysis: see Methods) support the inference from individual measures and datasets (Fig. 1A–E; see also Supplementary Fig. S4), with statistically significant ($p < 0.05$ [two-sided] via simultaneous confidence intervals) age-related differences detected throughout early to late adolescent periods (10–18 years old) for both accuracy and latency measures (Fig. 2E). While statistically significant ($p < 0.05$ [two-sided] via simultaneous confidence intervals) age-related differences could also be observed in this highly powered aggregate analysis until 20 years old for accuracy measures (Fig. 2E), the absolute magnitude of these effects were very small after 18 years old (mean z unit change in accuracy per-year between 18–20-years old: .023 [-1/5$^{th}$ the average change observed between 10- and 15 years old]); Fig. 2E). A parallel analysis examining the magnitude of change among those measures with statistically significant overall age effects (corrected $p$'s < 0.004, [two-sided]; see also Fig. 1, Supplementary Table S3) likewise demonstrates that, on average, over 95.0 and 99.7% of the total detectable age-related change between 8–35 years old occurs prior to 18 years old for accuracy and latency, respectively (Supplementary Fig. S5). These results provide robust and reproducible evidence of statistically significant and developmentally specific changes in executive function during early through mid-adolescence that reach maturity between 18 years old and 20 years old and reinforce that adolescence is a period of ongoing development of goal-directed cognition and executive function. A normative maturational stability towards adult-levels of executive function by late adolescence (18- to 20 years old) is highly consistent with what has been theorized in heuristic models of adolescence (~20 years old), but notably earlier than lifespan accounts suggesting executive function changes continue to occur during emerging adulthood (18–25 years old).

### Adolescent executive function development is predominantly domain general
Building from the observation that nearly all executive function measures showed the same developmental trajectory and relative maturational timing, we next examined the potential shared information across measures at the per-participant level using between-person (all datasets) and within-person (Luna, NCANDA) correlations and

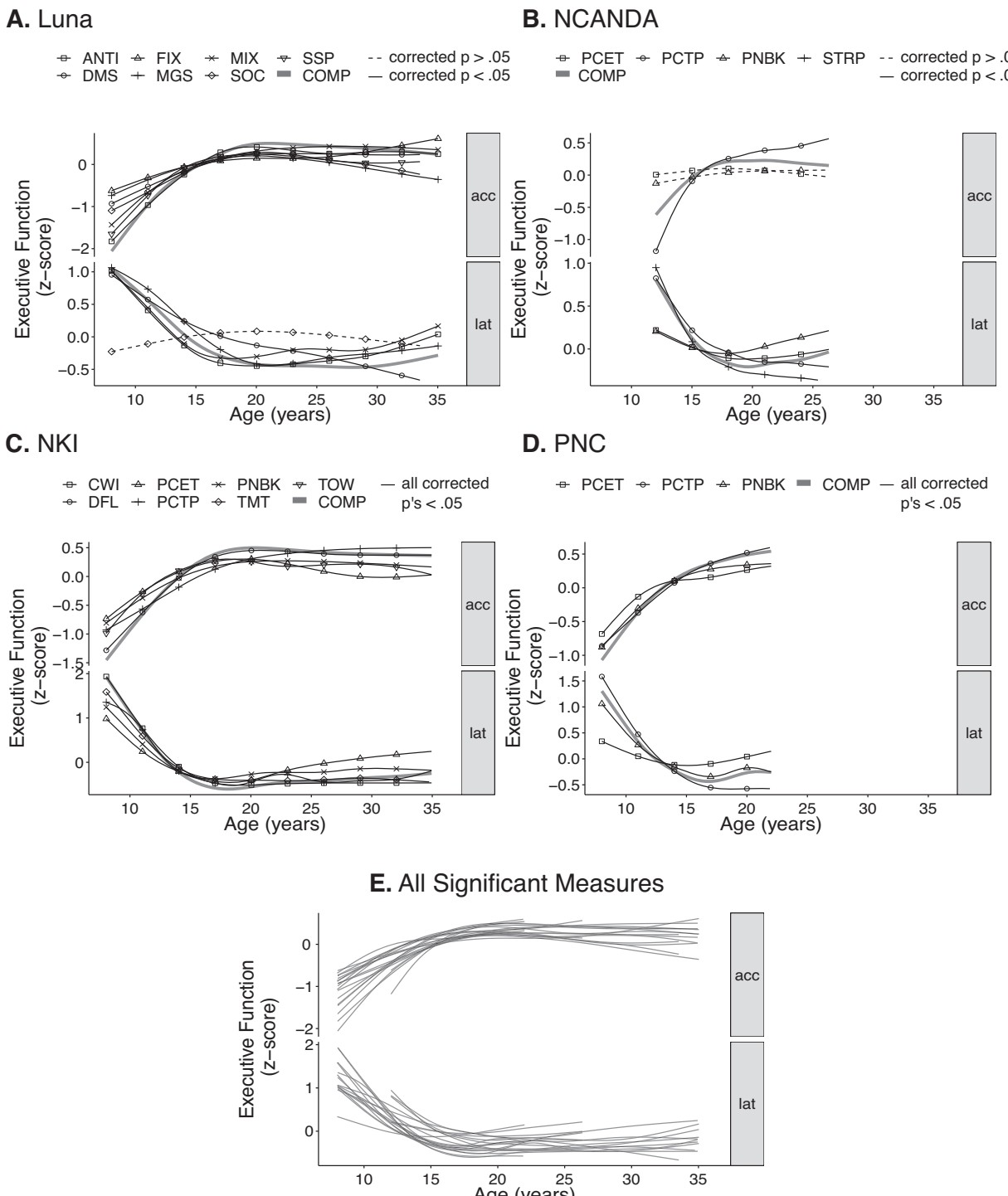

factor analysis (see Methods). Composite metrics were not used here, as they are by construction (linear sums of original measures) correlated with multiple measures. Consistent with a domain general, unity process of executive function, per-participant scores across nearly all measures were moderately correlated (see Methods) in all datasets in both between-person (cross-sectional) and within-person (longitudinal) analyses (Fig. 3A; mean linear, bivariate correlation from data aggregation ("all measures") $|r| = 0.261$; Supplementary Table S4 for correlation matrices). Exploratory factor analysis likewise demonstrated that a single domain general factor (via bifactor rotation) explains over 20% (21.9%) of total executive function variance on average, across datasets (Fig. 3B). There was no systematic evidence

that the total executive function variance explained by a single domain general factor varied by age (Supplementary Fig. S6). While certain data-driven thresholds to determine the number of supported latent factors (parallel analysis, optimal coordinate, acceleration factor, and a factor analytic Kaiser rule; see Methods, Supplementary Fig. S7) suggest the inclusion of a second or third factor across datasets (Fig. 3B), these factors account for very small amounts of executive function variance (on average, ~6 and 2% respectively, see Fig. 3B; Supplementary Fig. S7 for individual datasets) after accounting for the domain general factor (via bifactor rotation). Visual inspection of loadings for secondary and tertiary factors demonstrate that these factors tended to capture residual effects from specific, single measures or methods

**Fig. 1 | Age trajectories of executive function measures.** All measures scaled to per-dataset standard deviation ($z$) units. **A** Non-linear fits from the Luna dataset ($N = 196$; 666 total visits) of general additive mixed model (GAMM; multilevel penalized spline regression) for Antisaccade (ANTI), Fixation Breaks (FIX), Mixed Antisaccade/Visually Guided Saccade (MIX), Spatial Span (SSP), Delayed Matching to Sample (DMS), Memory Guided Saccade (MGS), Stockings of Cambridge (SOC), and equally weighted composite metrics ($z$ score sum of all accuracy, latency measures; COMP) as a function of age for accuracy measures (acc; top) and latency (lat; bottom). All models covaried for a smoothed effect of visit number. Solid line indicates models with Bonferroni corrected significance (corrected $p < 0.05$ [two-tailed], unadjusted $p < 0.003$ [two-tailed]). Dashed indicates models that do not surpass this threshold. **B** Non-linear, GAMM fits from NCANDA dataset ($N = 831$; 3412 total visits) for Penn Conditional Exclusion (PCET), Penn Continuous Performance (PCTP), Penn N Back (PNBK), and Stroop (STRP) tests and equally weighted accuracy, latency composite metrics (COMP) as a function of age. All models

covaried for a smoothed effect of visit number. Solid line indicates models with Bonferroni corrected significance (corrected $p < 0.05$ [two-tailed], unadjusted $p < 0.006$ [two-tailed]). Dashed indicates models that do not surpass this threshold. **C** Non-linear fits from NKI dataset ($N = 588$) of general additive models (GAM; penalized spline regression) for D-KEFS Color-Word Interference (CWI), Penn Conditional Exclusion (PCET), Penn N Back (PNBK), D-KEFS Tower (TOW), D-KEFS Design Fluency (DFL), DKEF Penn Continuous Performance (PCTP), and D-KEFS Trails (TMT) tests and equally weighted accuracy and latency composite metrics (COMP) as a function of age. All models were corrected, significant (corrected $p$'s $< 0.001$ [two-tailed]). **D** Non-linear, GAM fits from PNC dataset (9151) for Penn Conditional Exclusion (PCET), Penn Continuous Performance (PCTP), and Penn N Back (PNBK) and equally weighted accuracy and latency composite metrics (COMP) as a function of age. All models were corrected, significant (corrected $p$'s $< 0.001$ [two-tailed]). **E** Fits from all corrected significant models from **A–D**. See also Supplementary Table S3 for accompanying statistics.

(e.g., eye-tracking) or similar, broad domain general patterns (see Supplementary Fig. S7). Additional factors beyond these (4 or more factors) were not suggested for any dataset, under any data-driven threshold (Fig. 3B; Supplementary Fig. S7). Combined, these results provide evidence across studies for a single domain general factor or unity framework of executive function factor that accounts for variance across tasks (see Fig. 3C), although further work with expanded measures can help clarify potential diversity and domain-specific executive function performance (see Discussion).

Beyond the general dimensionality of participant-level, individual differences, a primary goal of the current work was to determine the timing and complexity of age-related differences in executive function from adolescence to adulthood. Therefore, we next tested the extent to which age-related, developmental differences in any one specific executive function measure could be explained by the general executive function processes supported in our previous analyses. Through nested model comparisons (see Methods), we determined the percentage of age-related differences on each specific executive function measure explained by a single domain general composite metric of the accuracy and latency metrics from the remaining tasks in the dataset ("leave one task out" composite metric; see Supplementary Fig. S8 for visualization of this procedure) versus a measure and/or task-specific process. As the broadest test of such a domain-general executive function process and consistent with prior suggestions from related literature in aging[52,53], in datasets (Luna, NKI) where multiple measures had the same putative, primary executive function subdomain (see first listed Domain in Supplementary Table S2), these measures were likewise left out of the composite metric ("leave out all measures from the same domain"). To further maximize comparability across studies and to prevent bias from shared, non-executive function visit effects (e.g., practice effects; see Sensitivity Analyses and Supplementary Fig. S9), analyses here were performed with the larger cross-sectional data, but were consistent with longitudinal data (cf., within-person factor structure in Fig. 3B, D).

Results demonstrated that a general component of executive function (as a single composite metric) often explained more than half of age-related information (via deviance testing in model comparison; see Supplementary Fig. S8) in individual executive function measures, with age effects for several measures nearly fully explained by a general executive function process (Fig. 4A–D). Aggregate analysis (three-level meta-analysis) revealed that on average, close to three-fourths (i.e., 75%) of age-related information in any one executive function measure could be explained by a domain-general executive function process (via a single composite metric of [equally weighted] out-of-domain measures; percentage of explained age-related deviance by common executive function for accuracy measures: 79.3%, latency measures: 70.6%; Fig. 4E). There was, however, notable variability between the proportion of explained variance by common executive function across datasets. One possible explanation for these

differences is that the datasets (NCANDA, PNC) with fewer executive function measures have less precision to estimate a domain general executive function process. Consistent with this, the percentage of age-related information explained by a common executive function process decreased and became more variable across measures in Luna and NKI datasets in simulations that used iteratively smaller numbers of variables to estimate an executive function composite (see Supplementary Fig. S10). Combined, these results provide the strongest evidence for a core domain general or unitary process related to observed age-related differences in executive function that is reproducible across measures and datasets. Together with our previous analyses, these results support adolescence as a potentially specific period of the lifespan of ongoing executive function, where a core unitary maturational process may give rise to improvements across related but distinct assessments.

## Scaled domain general executive function scores generate reproducible normative maturational templates across datasets and tasks

Having established that executive function measures follow a canonical developmental trajectory during adolescence and age-related changes are well captured by domain-general processes, our final analyses sought to build upon these results to create normative maturational templates applicable across datasets and tasks. That is, if a substantial portion of executive function development follows the same trajectory (Figs. 1, 2) and is driven by a common, domain-general process (Figs. 3, 4), we tested whether a simplified normative template of change would be representative across new datasets and tasks and could be used to quantitatively guide future research.

A standard growth chart[54] constructs a normative template of developmental change and inter-individual variability (e.g., percentile) for a single assessment with a single scale of measurement (e.g., height in inches). Executive function, however, is assessed with dozens of different measurements[5] and owing to the potential range of participant ages included in any one developmental dataset, the total extent of observed individual variability may substantially differ across datasets, even if developmental change proceeds according to the shape of the canonical executive function trajectory. In the current datasets, scaling to adult performance (standard deviation units based on performance of 20–30-year-olds in each dataset; see Methods) to approximate a common scale provides a further robust demonstration of the shape of the canonical executive function trajectory for domain general accuracy (Fig. 5A) and latency (Fig. 5B) across datasets and tasks (given each dataset includes different measures; see Supplementary Table S2). Differences in the precise scaling (absolute y values at each age) persist, however, as to be expected by datasets taken from different age ranges with different tasks. Furthermore, such universal scaling to adult performance, while potentially useful for creating a common metric across measures and tasks, would not be possible for

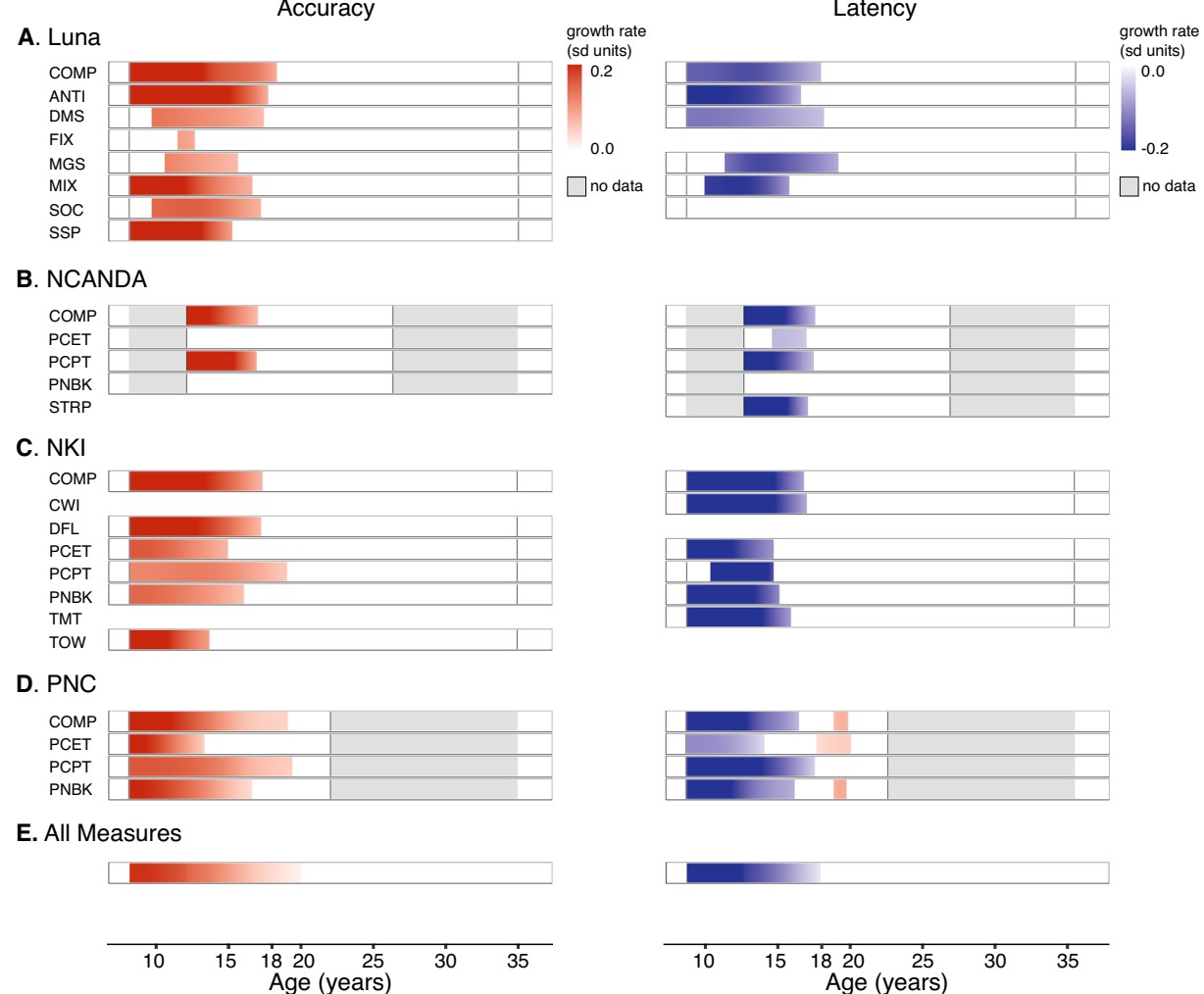

**Fig. 2 | Developmental periods with significant age-related change in executive function.** Age-ranges in which the simultaneous (to account for multiple testing) 95% confidence interval (generated via posterior simulation[50] with 10,000 iterations) of the first derivative of the GAM/GAMM fits did not include zero ($p < 0.05$, two-sided) were classified as statistically significant. Using this method, raster plots display color (red: age-related increases; blue: age-related decreases) when the derivative is statistically significant ($p < 0.05$, two-sided) and white when the derivative is not statistically significant ($p > 0.05$, two-sided). Vertical black lines in each bar denote the minimum and maximum age of the included dataset. Gray bars indicate no data within the specified age range for that analysis dataset. Accuracy measures for all datasets are shown on the left and latency on the right. **A** Luna dataset ($N = 196$; 666 total visits) measures from equally weighted composite metrics (z score sum of all accuracy, latency measures; COMP), Antisaccade (ANTI), Delayed Matching to Sample (DMS), Fixation Breaks (FIX), Memory Guided Saccade (MGS), Mixed Antisaccade/Visually Guided Saccades (MIX), Stockings of Cambridge (SOC), and Spatial Span (SSP). **B** NCANDA dataset ($N = 831$; 3412 total visits)

measures from equally weighted accuracy, latency composite metrics (COMP), Penn Conditional Exclusion (PCET), Penn Continuous Performance (PCTP), Penn N Back (PNBK), and Stroop (STRP). **C** NKI dataset ($N = 588$) measures from equally weighted accuracy and latency composite metrics (COMP), D-KEFS Color-Word Interference (CWI), D-KEFS Design Fluency (DFL), Penn Conditional Exclusion (PCET), Penn Continuous Performance (PCTP), Penn N Back (PNBK), D-KEFS Trails (TMT), and D-KEFS Tower (TOW). **D** PNC dataset ($N = 9151$) measures from equally weighted accuracy and latency composite metrics (COMP), Penn Conditional Exclusion (PCET), Penn Continuous Performance (PCTP), and Penn N Back (PNBK). **E** An aggregate analysis (pointwise three-level meta-analysis), incorporating all measures from all datasets was performed using the metafor package[51] with effects nested in measure and study (see also Methods) and thresholded in the same manner as the other bars (red or blue denoting statistically significant [$p < 0.05$, two-sided; via simultaneous confidence intervals] age-related increases or decreases, respectively).

future studies that only assessed a narrower age range (e.g., 10–18 years old).

We sought to establish a procedure for constructing normative maturational templates applicable to all relevant ages (8–35 years old) that utilizes a linear scaling of the canonical executive function trajectory to a specific measure (via basis function regression; see Methods). Unlike the GAM/GAMMs used to initially derive the canonical executive function trajectory or alternative, multiparameter nonlinear models of age, we tested a procedure that only requires a simple linear transformation of the age variable in each dataset (via linear interpolation to the canonical trajectory [estimated out-of-dataset];

see Methods) and is then fit as a single parameter in a general linear model/general linear mixed effects model. This data-driven basis function process (see analogous ideas in functional brain imaging[55]) is therefore the same as what occurs with standard parametric functional forms of age (e.g., linear, inverse linear age [1/age], quadratic polynomial age [age + age²]), but would have the added benefit of its shape/functional form being directly informed by prior developmental data. We tested this procedure to directly assess whether the insights generated in the current work regarding a canonical executive function trajectory could quantitatively guide future research allowing for simplified modeling approaches that are developmentally informed

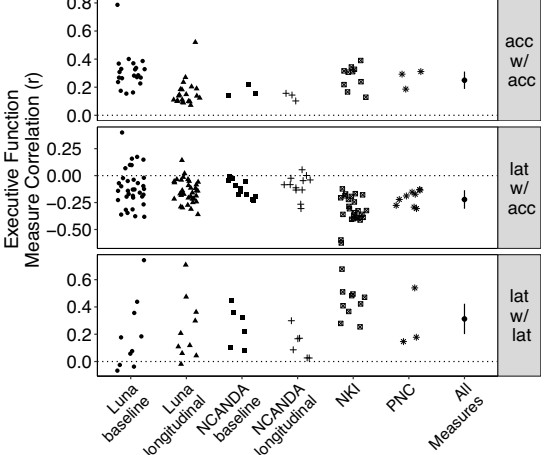

**A. Executive Function Measure Correlations**

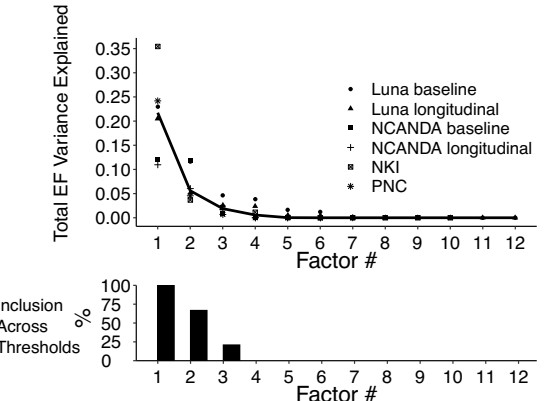

**B. Factor Analysis: Variance Explained, N Factors Supported**

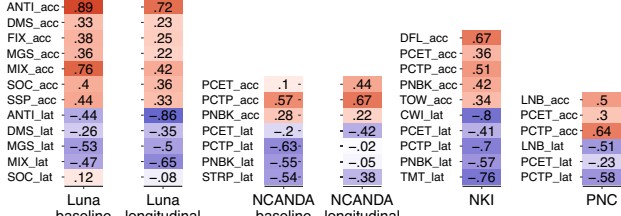

**C. Domain General (Factor 1) Loadings**

**Fig. 3 | Correlation and factor structure of executive function measures.**
**A** Linear, bivariate correlation (*r*) for Luna (*N* = 196; 666 total visits), NCANDA (*N* = 831; 3412 total visits), NKI (*N* = 588), and PNC (*N* = 9151) datasets among accuracy measures (acc w/ acc), accuracy with latency measures (acc w/ latency), and among latency measures (lat w/ lat). For longitudinal datasets (Luna, NCANDA), baseline correlations were calculated from first visit, longitudinal correlations were calculated from disaggregation analysis (see Methods). "All measures" indicates estimate (black dot; measure of center) and 95% confidence interval (± 2 standard errors) from three-level meta-analysis (correlation pairs nested in task pairs and datasets). **B** Top panel displays total executive function (EF) variance explained as a function of extracted factor using a bifactor rotation for each dataset (maximum number of factors extracted per dataset based on total measures per dataset [Luna, 12 measures/factors, NCANDA 7 measures/factors, NKI 10 measures/factors, PNC 6 measures/factors]. Black line indicates mean across datasets. Bottom panel displays factor inclusion across thresholds (parallel analysis, optimal coordinate, acceleration factor, and a factor analytic Kaiser rule; see Methods) and datasets (e.g., 100% indicates factor included across all thresholds and datasets; see Supplementary Fig. S7 for individual datasets). **C** Loadings for domain general factor (factor 1 via bifactor rotation) for each dataset and by baseline and longitudinal (via disaggregation; see Methods) for Luna and NCANDA.

and computationally efficient. To mirror the use of this approach in future developmental research with new datasets and new measures, we tested the generalizability of this procedure through cross-validation ("leave one dataset out") and compared performance to standard functional forms of age used in developmental research (linear age, inverse linear age [1/age], quadratic polynomial age [age + age$^2$]) that may otherwise be used to understand age-related executive function change and deviations from normative development.

A canonical executive function trajectory, estimated out-of-sample ("leave one dataset out") and used as a single parameter basis function (e.g., shape of age model for Luna dataset determined by NCANDA, NKI, and PNC datasets; see Supplementary Fig. S11 for visualization of workflow), generally outperformed standard functional forms of age (linear age, inverse linear age [1/age], quadratic polynomial age [age + age$^2$]) during model comparison testing that aggregated multiple metrics of model fit and complexity (Fig. 5C, D). Following model selection criteria based on all metrics across accuracy and latency measures, the simplified, single parameter basis function was the most selected model (55.6% of the time; compare to quadratic [age + age$^2$]: 37.3%; inverse linear age [1/age]: 7.03%; linear age: 0%), which was significantly higher than all other age models (vs. inverse linear age [1/age]: $\chi^2$ = 21.6, $p < 0.001$; vs. linear age $\chi^2$ = 30.6, $p < 0.001$; all $p$ values two-sided; chi-square test with Yate's correction for continuity) other than the quadratic model (vs. age + age$^2$ (best model 37.3%), $\chi^2$ = 2.29, $p$ [two-sided] = 0.130). Results were further unchanged when specifically looking at generalizability between Luna and NKI datasets that do not share any measures (data-driven age basis model was best age model overall 69.2%). Consistent with the strength of the basis function being derived from its developmentally precise shape, offsetting the basis function with respect to age led to lower and more variable model performance (Supplementary Fig. S12). Combined, these results establish a simplified, single parameter data-driven basis function version of the canonical executive function trajectory as an alternative, developmentally informed functional form of age that is superior or highly competitive with standard, parametric functional forms of age when applied to new datasets and new measures. Therefore, we suggest that, along with full, multi-parameter complex spline models (GAM/GAMMs used throughout the rest of the manuscript) and standard functional forms of age (e.g., linear, inverse linear, quadratic), such a simplified, developmentally informed basis function may quantitatively (see Data and Code Availability) inform future research on normative development and deviations from normative development in health and disease (see Discussion).

**Sensitivity analyses**

Sensitivity analyses demonstrated that primary results concerning the magnitude and timing of executive function accuracy and latency development were consistent across males and females (Supplementary S13). Additional sensitivity analyses demonstrated that our primary results did not change when covarying for socioeconomic indicators (parental education and family income, Supplementary S14, S15) and assessments of culturally acquired knowledge (verbal reasoning and vocabulary, Supplementary S16), and remained consistent across mental health inclusion/exclusion thresholds (Supplementary S17). This suggests that mathematically holding these factors constant did not change the current results that focused on aggregate and average executive function changes during adolescence. Thus, our results do not speak to for example past findings suggesting economic disparities impact cognitive measures, and variability between individuals (cf.,[56,57]). However, the tools and insights from the current work can be used for future studies focused on relationships between these factors and executive function in more detail (see Discussion). As in previous longitudinal investigations of computerized and neuropsychological performance[58], age-independent visit effects (e.g. practice effects) on cognitive testing were observed for many

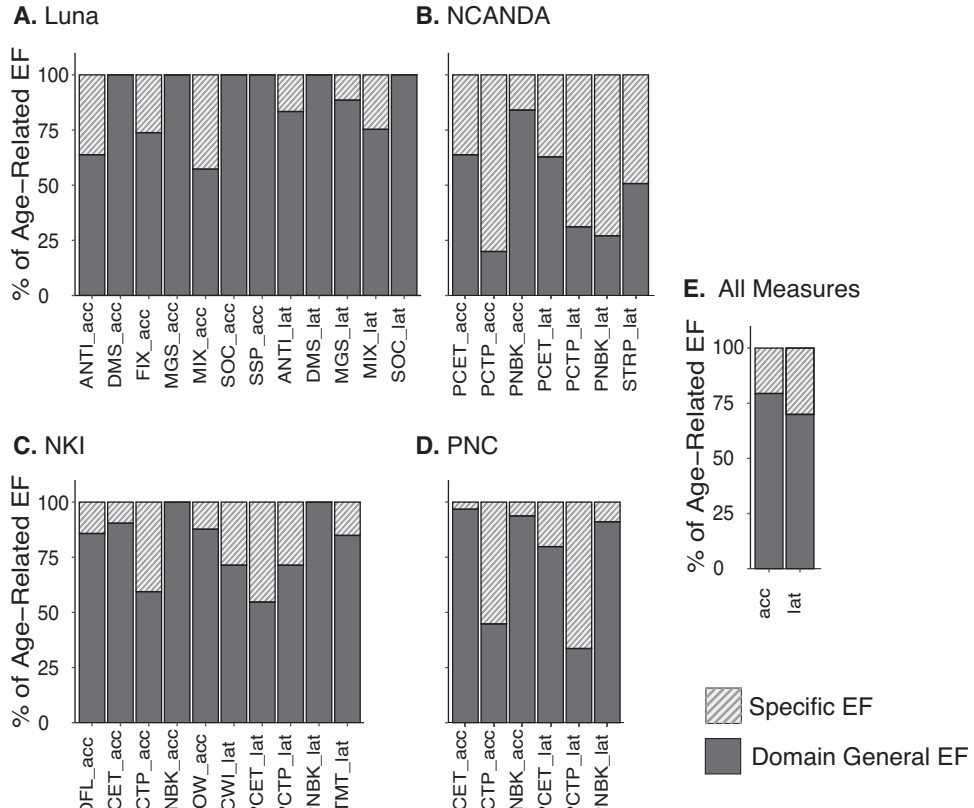

**Fig. 4 | Contributions from domain-general versus specific processes to age-related differences in executive function.** For each measure in each dataset (**A** Luna; **B** NCANDA; **C** NKI; **D** PNC), model (GAM) comparisons were used to identify the percentage of age-related deviance attributed to a measure and domain-specific process (hashed bars) from that of a domain-general process (filled bars). Specific effects represent the incremental deviance attributed to that measure over a single, equally weighted composite metric of measures from the held-out domain (e.g., for inhibition and inhibition/switching measures in the Luna sample [Antisaccade, Fixation, Mixed Antisaccade] the composite metric was composed of working memory and planning measures; see Supplementary Fig. S8 for workflow visualization, Methods). Through model comparison, nonspecific effects are attributed to domain general effects. **E** All Measures effects are estimated via three-level meta-analysis nesting effects in measures and datasets.

executive function tasks in longitudinal samples (Supplementary S9). However, all longitudinal analyses (Luna, NCANDA samples) covaried for a non-linear effect of visit number (see Methods) and we demonstrate replication to two cross-sectional datasets (NKI, PNC) where visit effects could not have occurred, indicating that our primary results are likewise robust to practice effects on cognitive testing.

## Discussion

### Defining the adolescent period through a reproducible, canonical trajectory of executive function and significant periods of development

The development of executive function has been studied in relatively small (N's ~200[15,16]) independent investigations using a broad range of tasks or in relatively large studies (N's ~ 1000[17,18]; although still smaller than the total sample used here: $N = 10,766$, total visits =13,817)) with few, very narrow assessments of executive function in intelligence testing. Collectively, prior work demonstrates significant improvements from childhood through adolescence[5,15,16,19–24], but the precise magnitude, maturational timing, and significant periods of development in executive function during the transition from adolescence to adulthood has not been defined. With four, large independent datasets, and non-linear modeling techniques to identify specific periods of significant development, we provide reproducible and direct evidence that executive functions continue to develop into late adolescence, which has been widely suggested by theory[4,7–9] but has rarely been directly tested in empirical research. Building from prominent neurodevelopmental[4,7–9] and psychological[27,28] theories, these results

highlight adolescence as an essential period of transition during which individuals reach maturity in goal-directed cognition. This suggests that while adolescents clearly possess complex cognitive abilities, including the ability to inhibit prepotent responses, maintain and update information in memory, and abstractly plan for future events, such abilities do not reach their full potential until 18–20 years old (late adolescence). Adolescent periods prior to this age-range (i.e., early to mid-adolescence ~10–15 years old, and mid to late adolescence ~15–18-years old) are therefore likely critical final stages of this type of cognitive development, where deviations from normative development may lead to poorer outcomes in adulthood. Identifying these sensitive, or even critical[3], periods of cognitive development is essential for advancing neurocognitive growth-charting to determine normative development and deviations from this normative development in health and disease[19,45], in designing developmentally informed interventions/preventions for youth[59–62], and policy concerning adolescents[9,14].

Given the reproducible and converging evidence for adolescence as a distinct period of the lifespan, and one now better conceptualized as a period of normative closure in goal-directed cognitive development prior to the establishment of adult-level trajectories, the current results support a broader understanding of the neurobehavioral basis for the adolescent period. Together with essential additional historical and sociocultural frameworks[27], such charting of neurobehavioral processes throughout adolescence emphasize the importance of developmentally relevant considerations for adolescents across research and clinical care. Thus, our identification of the maturational

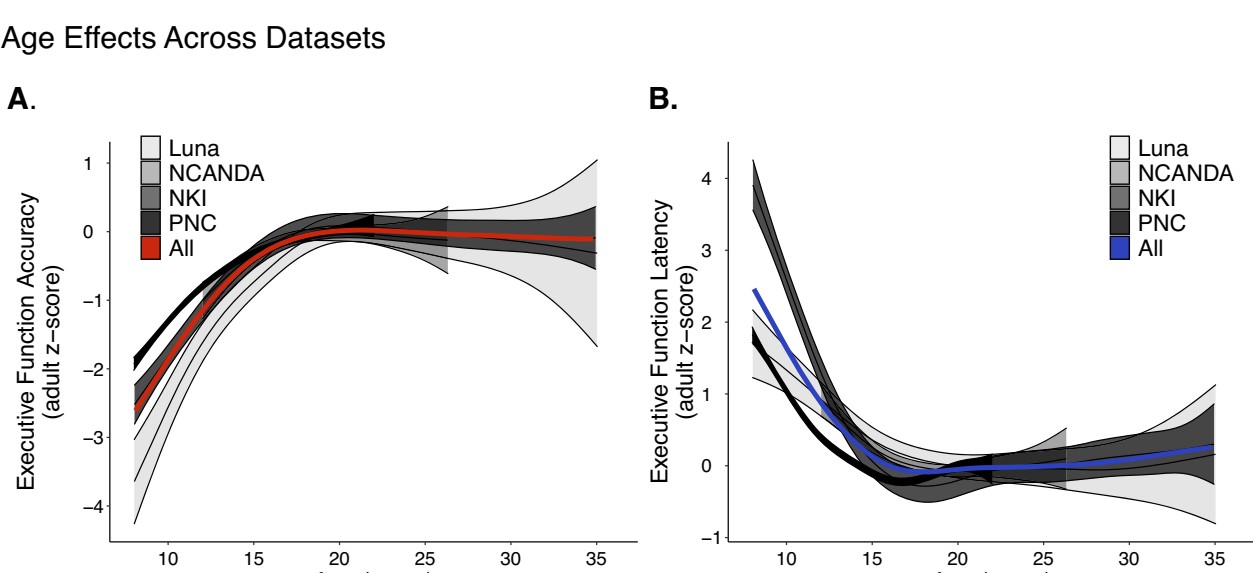

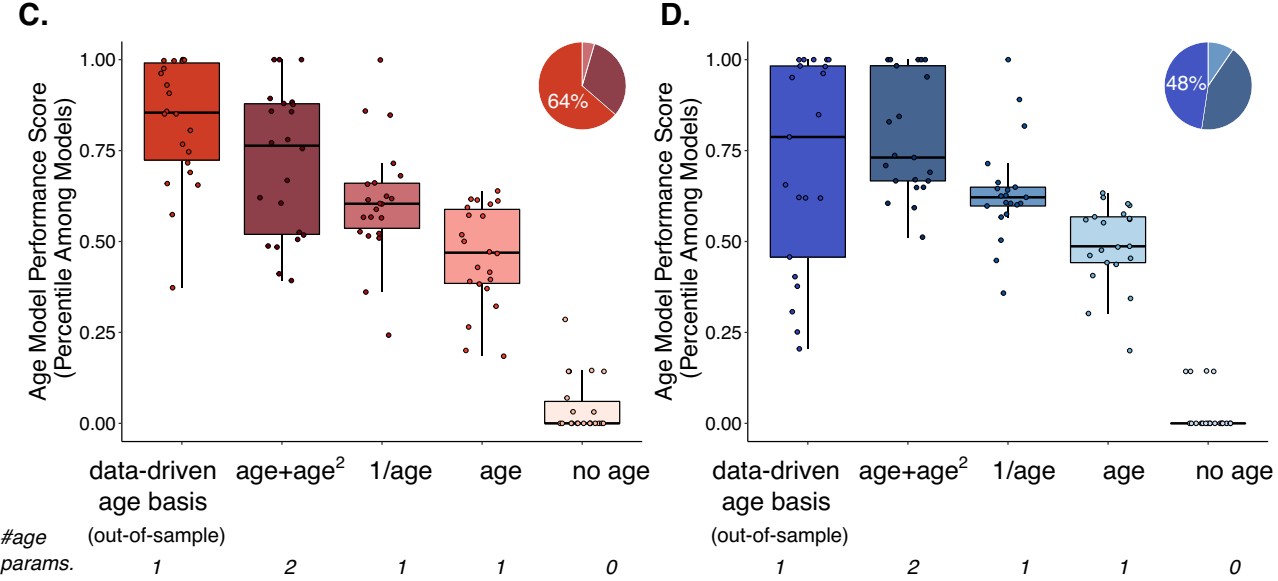

**Fig. 5 | Scaled domain general executive function scores generate reproducible adolescent growth charts across datasets and tasks.** Accuracy (**A**) and latency (**B**) composite (z score sum of all accuracy, latency measures; see Supplementary Table S2) executive function scores for Luna ($N = 196$; 666 total visits), NCANDA ($N = 831$; 3412 total visits), NKI ($N = 588$), and PNC ($N = 9151$) datasets. Each measure within each dataset is $z$ scored to the performance of adults (participants 20–30 years old). Fit lines are from GAM/GAMM models. Error bars represent two times the standard error added above and below these fits (measure of center). **C, D** Out-of-dataset performance as boxplots (center line, median; box limits, upper and lower quartiles; whiskers, 1.5x interquartile range; points, outliers) of the single parameter data-driven age basis function for accuracy (**C**) and latency (**D**) measures relative to typical age models (quadratic [age+age$^2$], inverse age [1/age], linear age [age]) and an intercept only (no age) model. One dot per measure from all datasets ($N = 22$ accuracy; $N = 21$ latency). Cross-validation ("leave one dataset out") was used to validate the age basis function derived from the canonical executive trajectory. See Supplementary Fig. S11 for diagram of procedure. Potential age models were evaluated with multiple metrics of model fit and complexity (see Methods and Supplementary Fig. S11). Using the performance package (rank function) in R[55], model fit metrics were scaled 0 (worst model on that fit metric) to 1 (best model on that fit metric, accounting for the directionality of improved fit for each metric [e.g., $R^2$ larger values, RMSE lower values]) across candidate age models and the mean value across all model fit metrics was taken for each candidate age model to create an overall performance score (y-axis; **C, D**). Pie charts indicate the percent of times that each age model was the top ranked according to this procedure; color in pie chart corresponds to age models color from boxplots. Number of age parameters (# age params.) specifies the number age variables used in each candidate age model (see also Supplementary Fig. S11).

timing of executive function, in combination with similar investigations of affective and social processes[2,63] may guide further discussion on how to define the adolescent period and demarcate its boundaries[27], essential for basic and translational developmental research. To assist in this pursuit, we have made available summary data (note participant-level data is also available with necessary data use agreements; see Data availability) for the canonical executive function trajectory, with the goal that subsequent work may utilize and

continue to refine empirically defined normative maturational templates in executive function research. While such refinement should include ongoing model comparison of other candidate functional forms of age (cf., Fig. 5), we suggest sharing of reproducible and well-powered adolescent trajectories of executive function can be directly integrated in future analysis (e.g., basis function regression) in methods that mirror the development, refinement, and use of summary statistics in other fields (e.g., polygenic risk scores[64]). As in these and related fields[65], large-scale reproducible normative templates of change can be leveraged to better understand risk factors or consequences of mental and physical health conditions related to executive function during adolescence and across a range of experimental conditions.

Three of the four datasets used here (NCANDA, NKI, PNC), as community samples, did not exclude participants on the basis of mental health presentations. However, our sensitivity analyses demonstrated that our approach (used in an effort to maximize generalizability; c.f.,[66,67]) did not bias our results that focused on aggregate and average executive function changes during adolescence. The tools and insights developed here can support future studies of executive function differences in psychopathology both in new datasets, as well as targeted investigations within the current datasets. Normative templates of age-related differences in executive function derived here may also be useful for future research to disambiguate developmental effects and non-developmental visit effects (e.g., practice effects) that, consistent with prior reports[58], we observed in longitudinal executive function data. Future work may also use these insights towards optimizing developmental study designs with respect to the number of participants, construct breadth of assessments, and the number of longitudinal time points.

The results of the current work provide support for prior theoretical and quantitative work suggesting non-linear developmental trajectories of cognition during adolescence[15,20,32]. Updating theoretical models requires broad conceptual consideration, nevertheless, the clear presence of non-linearity in age-related executive function differences from late childhood through adulthood can directly help refine neurodevelopmental models of adolescence. Our results for example provide less support for linear increases of executive function development throughout adolescence[8] as well as maturational timing of this process after twenty years old[27]. Instead, our results clearly support a reproducible, canonical non-linear trajectory of executive function development from adolescence to adulthood. The shape and timing of this canonical trajectory is consistent with prior theories of adolescence[9] and empirical work with fewer executive function assessments and/or smaller samples that suggest non-linear cognitive development processes[17,32]. The robust, large-scale multi-dataset replication here provides key advances towards formalizing such a non-linear trajectory, and through the employed data-driven modeling approaches, explicitly defines significant periods of executive function development that identify the potential closing of the adolescent period for this process between 18–20 years old. Such distinctions on the relative bounds of the adolescent period are not only essential for psychological and neuroscientific theories, but also for clinical care and policy. Our work also sets key areas for future work regarding maturational timing in more fundamental studies of executive function development (e.g., disambiguating age-related change from pubertal development, generalizability to populations outside of the United States, targets for brain imaging, and considerations for affective versus nonaffective executive function tasks: see Considerations for Future Work).

## Domain general executive function development

While prior work in adults[52,68] and younger children[69–71] has provided evidence for a potential unity/diversity framework of executive function, the relative domain-generality versus specificity of executive function has largely not been examined with respect to changes during adolescence. The strongest evidence across the large-scale data aggregated here suggests that age-related differences and longitudinal changes across executive function tasks are driven predominantly by a domain-general process. This indicates that across executive processes (e.g., inhibitory control, attention, working memory, planning) there is a common system of goal-directed cognition that may lead to developmental improvements across multiple contexts. Such domain-general executive function development may help explain, for example, wide-spread differences across executive function tasks in clinical[72,73] and/or population research (e.g., social determinants of health[71,74,75]), as well as the tendency for many executive function tasks to engage common neural circuitry[76,77]. Domain-general executive function development during adolescence also provides support for general heuristic perspectives of adolescence that emphasize a core set of cognitive development[4,7–9]. The current work that focused on multi-assessment and multi-dataset reproducibility of trajectories of adolescent executive function across large-scale cross-sectional and longitudinal data further sets priorities for additional within-person modeling (e.g., multivariate sparse functional principal components analysis[78], multivariate growth curve modeling[79]) in future targeted investigations.

Although we found a considerable degree of commonality in adolescent executive function development, as in related work from adults[42], current measures and methods do not rule out additional executive function variance relevant to development (even if such domain/measure-specific variance is less prominent than domain-general processes). Our analyses were generally well accounted for by a domain general perspective of executive function, and further exploring this allowed us to examine multi-assessment multi-dataset estimates towards reproducibility and generalizability. However, as in other reports[42,69–71], executive function variance was not entirely captured by a single factor. Future work, including using the tools and insights developed here, may address these questions in multi-dataset reproducibility and generalizability investigations. With respect to potential distinction among other cognitive processes, our sensitivity analyses did however demonstrate that the canonical executive functional trajectory was robust to individual differences in measures of culturally acquired knowledge (see Supplementary S16). The results here nevertheless raise further questions regarding the conceptual distinction of executive function performance and development from that of related domain-general concepts like fluid cognition that are theorized to account for the coherence of performance-based cognitive abilities (and the distinction from culturally acquired knowledge) in the context of general ability testing (see[80] for additional discussion). Future empirical and theoretical work, to add to existing frameworks, will be required to rectify these related but often historically distinct accounts. From either account, we suggest that commonality across measures, while essential for basic and translational research and practical demarcations of adolescent development, be expanded to consider broader sociocultural and historical perspectives as well. The increasing availability of future large-scale population-level cohorts (e.g., Adolescent Cognitive Brain Development [ABCD] Study[81]), together with the methods used and developed in the current work, can facilitate future empirical investigations into these areas.

## Common driver of executive function development

Conceptually, the potential cognitive and psychological mechanisms of such domain general executive function development remain somewhat of an open question. As inhibitory control tasks (anti-saccade, color-word interference, trail-making-test) often had both the highest loadings on domain-general factors observed here (which is consistent with similar prior work in adults[68]) and amongst the largest developmental effects, it is possible a global inhibitory control process provides the most parsimonious explanation for domain-general/

unitary executive function. If, as has been suggested, executive function tasks often fail to solely isolate a specific cognitive process (the so-called task "impurity problem"[5,68,82]), global inhibitory control processes may give rise to broad executive function changes through adolescence across diverse tasks, each of which requires some level of global, goal-directed inhibition. Nevertheless, we suggest that future work determining the common driver of executive function changes will benefit most from novel dense longitudinal study designs (e.g., repeated ambulatory smartphone/web-based assessment of cognition[83]) and/or further multi-method investigations (e.g., fMRI[76]) that provide a means to understand temporal processes and/or correlated neurobiology, respectively. This would help protect against the possible circularity of descriptions of a common driver of executive function that are limited to functions assessed contemporaneously and/or with the same methodology. As demonstrated in the current work, however, even without a clear narrative description of the origins of domain-general executive function, the maturation of domain-general executive function provides a means to qualitatively understand the adolescent period and quantitatively guide future work. In pursuit of these goals, the current results emphasize the utility of research designs that include not just large sample sizes and/or longitudinal data, but also multiple measures within a broader construct (executive function). Our results suggest the utility of shared information and/or the potential utility of convergent validity from multiple executive function indicators in outcome research when such construct depth is available. Even when more domain-specific effects are of interest, our results suggest that the estimation of domain-general executive function via a broad battery is optimal, as developmental differences on nearly all measures had sizeable influences from a more general process.

## Considerations for future work

The identification of common adolescent executive function development may guide future translational and multidisciplinary research. For example, our results suggest that neuroimaging research of adolescent executive function may be well-suited by leveraging multiple executive function tasks to examine shared information in association with brain structure/function or to better isolate domain-specific effects. Likewise, as has become increasingly common[72], translational research aiming to uncover adolescent executive function as a possible predictor or consequence of clinical presentations and/or as a target for intervention, may be best suited to approach executive function from a unitary, domain-general process that follows the canonical executive function trajectory revealed here. Methodologically, common metrics of domain-general executive function, and normative templates of change (even in scaled units: basis functions) may serve to increase reproducibility by facilitating overlap and replication efforts across instruments and datasets.

The current project leveraged multiple large independent datasets, developed methodological improvements permitting the identification of maturational timing of executive function, and investigated both common and specific components of executive function processes, but nevertheless potential limitations and explicit suggestions for future work should be considered. First, although this investigation used a comprehensive approach to characterizing executive function, these analyses focused on the most prominent outcome measures from these tests. This approach had the advantage of aligning the current analyses with predominant practices in the literature and the level of granularity supported by large-scale, public datasets, but future work would benefit from alternative and/or model-based, computational parameterizations of behavioral performance[84,85]. Furthermore, the breadth of executive functions indexed by these tasks was not exhaustive, and other domains of individual differences in cognition were not explored. For example, by design, this study, and many of the original datasets, did not examine executive function tasks

in the context of affective stimuli. That is, the included measures focused on what have been considered affectively neutral cognitive measures. This allowed us to specifically isolate fundamental properties of executive function development as typically understood, but future work with more diverse cognitive batteries should examine whether affective manipulations likewise follow the canonical executive function trajectory established here. Another potential limitation is that the current work did not try to disambiguate age-related changes from pubertal development, given challenges in independently estimating these effects in the presence of large cross-sectional age effects (cf.,[86]). However, it will be important for future work, particularly when focusing on early periods of adolescence to likewise seek large-scale multi-assessment, multi-dataset reproducibility for the specific role of pubertal status in driving executive function development. A further potential limitation arises from our general focus on the average executive function trajectory during adolescence. While we determined that our results were generally robust to multiple participant-level factors, the results of the current work should be interpreted as a normative template and individual and dataset-level variability is expected. Relatedly, while the aggregated datasets and inferences drawn here appear to approximate population patterns from the United States, further work with multinational and multicultural samples is required to determine the generalizability of these results to other countries and cultures. The tools and data developed here can nevertheless provide resources for additional research on deviations from this normative trajectory, promote improved estimates of uncertainty, and ultimately support potential translational efforts seeking to identify clinically relevant executive function-related processes during adolescence.

## Methods
### Participants

Data for this project were provided from participants of four existing projects (all with publicly available data). One internal dataset (Luna Dataset) and three external datasets (National Consortium on Alcohol & Neurodevelopment in Adolescence[36] [NCANDA], Nathan Kline Institute-Rockland Sample[35] [NKI], Philadelphia Neurodevelopmental Cohort[37] [PNC]) were included based on (1) their inclusion of executive function tasks performed in a developmental or lifespan dataset spanning the entirety of the adolescent period and (2) to aggregate the largest possible dataset to explore the aims of this project. The primary focus of the current work was on the adolescent period. To explicitly capture transitions into and out of adolescence as well as the entire adolescent period[33], we included participants ranging from late childhood to adulthood (8–35 years old). Lower (8 years old) and upper (35 years old) age ranges were selected to be as inclusive as possible, given the overarching goal of capturing non-linear developmental trajectories, while also ensuring that at least two separate datasets had participants in each age range. This meant that only participants from 8–35 years old from the NKI lifespan dataset were included (Full NKI Rockland Sample Range: 6–85). No participants were excluded based on age from the other datasets (Luna, NCANDA, PNC), which were designed to assess childhood to adolescence/adulthood and fully fell within this age range. In order to maximize generalizability and representation within the datasets (see refs. 66,67 for relevant discussion concerning neurodevelopmental studies), no other participant-level demographic exclusion criteria were applied to the datasets. Instead, we thoroughly examined the potential impact of such factors in a series of sensitivity analyses (see Supplementary Figs. S2; S13-S17).

One dataset was drawn from Dr. Beatriz Luna's longitudinal study of neurocognitive development (Luna Dataset). From this dataset, the current project included 196 participants (baseline age-range: 8–30 years old; 101 female participants, 92 male, 2 participants both sexes were reported, 1 participant unknown/not reported) dataset in an

accelerated longitudinal/cohort sequential design, with participants completing a range of follow-up visits (total participant visits = 666, median number of visits per-participant = 3; range of visits per-participant = 1–10; median months between visits = 13.3; range of months between visits = 5.97–81.73; see Supplementary Fig. S1 for graphical depiction of dataset by visit structure). Exclusion criteria for this dataset were medical conditions or medications known to affect eye movements and a history of psychiatric disorders, developmental cognitive disorders, or learning disabilities, in either the participant or a first-degree relative, and IQ scores at baseline below 80. Participants were recruited from the community surrounding the University of Pittsburgh Medical Center.

The second dataset was drawn from the multi-site, National Consortium on Alcohol & Neurodevelopment in Adolescence (NCANDA) (see ref. 36 for detailed sampling strategy and recruitment information). The current project used data from 831 participants (baseline age-range:12–21 years old, 423 female participants, 408 male) in the first five visits of the accelerated longitudinal design (total participant visits = 3412, median number of visits per-participant = 5; range of visits per-participant = 1–5; median months between visits = 12.17; range of months between visits = 4.98–23.97; see Supplementary Fig. S1 for graphical depiction of dataset age by visit structure). Exclusion criteria for NCANDA were Magnetic Resonance Imaging (MRI) contraindications (e.g., claustrophobia, non-removable metal in the body), head injury with a significant loss of consciousness, psychiatric disorders that might influence study completion (e.g., psychosis), and psychiatric medication (see[36]). A central goal of the NCANDA study was to examine the transition to significant substance use during adolescence and as a result, approximately 50% of the dataset was recruited based on subclinical factors thought to increase the likelihood of alcohol use disorder (AUD; see ref. 36). The inclusion of participants with psychiatric conditions however was shown to not substantively influence the current projects' analyses through sensitivity analyses (see Supplementary Fig. S17).

The third dataset was drawn from the lifespan Nathan Kline Institute-Rockland Sample (NKI)(see ref. 35 for detailed sampling strategy and recruitment information). The current project used data from 588 participants (age range of participants within the included dataset [see above for age rationale]: 8–35 years old; 284 female participants, 304 male). The NKI-Rockland Sample includes longitudinal follow-up data (up to two visits) on the included tasks here for a very small number of participants ($n = 10$) within our specified age range. However, given this represented such a small percentage of participants (<2% of dataset) and only included two visits, the current analyses only included the first visit from these participants and thus this dataset was utilized as cross-sectional (see Supplementary Fig. S1 for histogram of included ages). The NKI-Rockland Sample was recruited to match the ethnic and economic demographics of Rockland County, New York. Consistent with the community sampling approach, a moderate number of participants in the NKI dataset used here ($n = 286$) met criteria (DSM-IV TR) for at least one lifetime diagnosis of a psychiatric disorder. These factors were shown to not substantively influence the current projects' analyses that focused on average and aggregate developmental changes in executive function through sensitivity analyses (see Supplementary Fig. S17).

The fourth dataset was drawn from the Philadelphia Neurodevelopmental Cohort (PNC) (see ref. 37 for detailed sampling strategy and recruitment information). The current project utilized data from 9151 participants in the cross-sectional, PNC dataset (age range: 8–22 years old; 4753 female participants, 4365 male, 19 participants both sexes were reported, 14 participants unknown/not reported; see Supplementary Fig. S1 for histogram of ages). Exclusion criteria for PNC were being non-ambulatory and not in stable health, non-proficiency in English, physical and cognitive challenges in participation in interviews and neurocognitive assessment, and the presence of a disorder that

impaired cognition or motility (see ref. 37 for detailed inclusion information). Given the large community-based sampling procedure of the PNC, this dataset included participants with psychiatric disorders that may be associated with neurocognitive performance. The current project followed previous work with this dataset[87] regarding data inclusion (see below) and sensitivity analyses examined the influence of these participants on the current project's analyses (see Supplementary Fig. S17).

In all four datasets, research protocols were approved by the relevant institutional review boards (Luna Dataset: University of Pittsburgh; NCANDA: Duke University, Oregon Health and Sciences University, SRI International, University of Pittsburgh, University of California San Diego; NKI: Nathan Kline Institute; PNC: The University of Pennsylvania and Children's Hospital of Philadelphia) and participants over 18 provided informed consent, while participants younger than 18 provided written assent and parental consent. To our knowledge, no participant was involved in more than one of the studies. For the current analyses, no statistical method was used to predetermine the included sample size. All four datasets were included in their entirety, apart from analysis-specific exclusions detailed below (Data Processing). As observational studies, the included experiments were not randomized. Likewise, no blinding procedures were employed.

## Executive function measures

Data from Luna, NCANDA, NKI, and PNC datasets were used in the current project based on their inclusion of executive function tasks performed in a developmental or lifespan dataset that spanned the adolescent period. Classification of executive function tasks was based on prior theoretical[5] and empirical work[15,34,42,88], with a general operationalization of goal-directed cognitive behaviors that encompassed processes of inhibition, attention, working memory, switching, or planning. Where possible, prior work with the included tasks and datasets and/or test authors[34] was used to define whether specific tasks indexed executive function. To avoid potential influences of verbal skills potentially related to educational attainment, measures relying heavily on reading and language skills were not included (e.g., DKEFS-Twenty Questions, DKEFS-Proverb Test) as primary executive function assessments, but the influence of culturally acquired knowledge was shown to not influence primary results in a sensitivity analysis (Supplementary Fig. S16). Wherever possible, both accuracy and latency measures were selected, except when precedence from research or clinical assessment was clear on a predominant use of accuracy (e.g., DKEFS Tower) or latency (e.g., DKEFS Trail Making Test) measures owing to nearly universal ceiling/floor performance of the corresponding accuracy/latency measure and/or the corresponding measure was not collected/available. See Supplementary Table S2 for the conceptualized subdomains of the included executive function tasks based on author consensus and original test descriptions. See Supplementary Table S3 for reproducible variable names for public datasets (NCANDA, NKI, PNC).

Based on the above criteria, the Luna dataset included twelve measures from six executive function tasks that were completed at each visit: Antisaccade (ANTI), Memory Guided Saccade (MGS), a mixed (MIX) Antisaccade/Visually Guided Saccade/Fixation task, Cambridge Neuropsychological Test Automated Battery [CANTAB] Delayed Matching to Sample (DMS), CANTAB Spatial Span (SSP), CANTAB Stockings of Cambridge (SOC). Each of these tasks have been described in detail elsewhere (see for example, refs. 15,44). Scoring procedures and outcome measures were based on previous work from our group and general use in the literature. Briefly, the Antisaccade task required participants to inhibit a proponent response (saccade) to a peripheral stimulus (in four possible locations along the horizontal meridian) and saccade towards the opposite hemifield. Both accuracy (correct response rate across trials) and latency (median speed of antisaccades on correct trials) of the Antisaccade task were examined.

A second mixed version of the Antisaccade task was also performed, where participants performed an antisaccade but trials with different task demands were also interleaved. Specifically, in 1/3rd of trials, participants were required to saccade towards the peripheral stimulus (visually guided saccade) or in 1/3rd number of trials, simply maintain fixation. Both accuracy and latency of this mixed version were examined, but only calculated for the antisaccade trials (with the same scoring procedure as above), given the visually guided saccade is not thought to rely on executive function (see ref. 15) and the number of fixation errors was included in a different measure that captured this performance in a goal-oriented context (see below). The Memory Guided Saccade task required participants to saccade towards a peripheral stimulus (in four possible locations along the horizontal meridian), remember its location during a subsequent fixation period, and then saccade towards the remembered location when no stimulus was presented. Both accuracy (difference in degrees between initial saccade and the most precise saccade the final phase[85], when no stimulus was presented) and latency (median speed of the initial saccade during the final phase across trials[85]) of the Memory Guided Saccade task were examined. We also calculated the number of fixation breaks (FIX) during the middle phase of the memory guided saccade task as a putative measure of inhibition. In addition to the three eye movement tasks, the Luna dataset also included the Delayed Matching to Sample, Spatial Span, and Stockings of Cambridge tasks from the CANTAB Battery, each of which have been broadly used and whose stimuli can be found online (see www.cambridgecognition.com/cantab/). Standard accuracy (Delayed Matching to Sample: Percent Correct; Spatial Span: Span Length; Stockings of Cambridge: Problems Solved in Minimum Moves) and latency (Delayed Matching to Sampe: Median Correct Latency; Stockings of Cambridge: Mean Initial Thinking Time) measures from each of the three CANTAB tasks were examined. For interpretive consistency across measures in the Luna dataset, the direction of the scoring of two accuracy measures (Memory Guided Saccade inaccuracy [see above]; Number of Fixation Breaks) was multiplied by −1 to ensure that higher scores indexed better performance on all accuracy measures.

The NCANDA, PNC, and NKI datasets used versions of the University of Pennsylvania Computerized Neurocognitive Battery (CNB; https://webcnp.med.upenn.edu/). The current project utilized data from three CNB tasks that met our operationalization of executive function and have been classified as executive by the CNB authors[34], the Penn Conditional Exclusion Test (PCET), a Penn N-Back Test (PNBK; NCANDA: Penn Short Fractal N-back Test [PNB-F]; PNC & NKI: Penn Letter N-Back Test [PNB-L]), and the Penn Continuous Performance Test: Number and Letter version (PCPT). Standard outcome measures for each task were included for accuracy (PCET: calculated accuracy measure [PCET ACC2]; PNB-F: true positive [correct] responses for 1-back and 2-back trials; PNB-L: true positive [correct] responses for 1-back and 2-back trials; PCPT: sum of true positives for number and letter trials) and latency (PCET: median response time for correct responses; PNB-F: mean of median response time for 1-back and 2-back trials; PNB-L: mean of median responses for 1-back and 2-back trials, PCPT: median response time for correct response to number trials and letter trials). The NCANDA dataset also included a standard Stroop Test (STRP), where the primary measure of average latency over all correct trials was included. The NKI dataset also included four executive function tasks from the Delis-Kaplan Executive Function System[43] (D-KEFS) that were included in the current study: color-word interference (CWI), design fluency (DFL), tower (TOW), and the trail-making test (TMT). Again, standard outcome measures were used for these tasks (CWI latency: average of inhibition and inhibition/switching conditions; correlation amongst these measures: $r = 0.806$; DFL Accuracy[88,89]: switching total correct; TOW: Total Achievement Score Total Raw; TMT: Number-Letter Switching). The DKEFS Sort Task was also available for a small percentage of participant visits within our

analytic age range (8–35) for the NKI dataset but was not used because over two-thirds of the visits did not have this measure (66.82%), whereas all other NKI measures included had at maximum <4% missingness.

## Data processing
All data processing and statistical analyses were performed in R version 4.1.2 (2021)[90]. Luna dataset eye-tracking data was scored with the same automatic scoring algorithms from our previous work[85,91]. Scores for all other tasks were generated through released software from the instrument (e.g., Luna dataset CANTAB) and/or included in official data releases (NCANDA, NKI, PNC datasets).

Aggregated data, either from distributed data releases (NCANDA, NKI, PNC) or our in-house database (Luna dataset) were first screened to ensure each visit (participant at testing session) had a valid age, anonymous id variable, and if longitudinal data, visit (i.e., these variables were not missing and were within the expected range, based on the study design) and included expected data. Data that did not meet these minimum criteria were removed from all analyses. As in our prior work, eye-tracking tasks in the Luna dataset (specific task at specific visit) with more than 30% of trials dropped due to poor eye-tracking or missing (i.e., early session termination; cf.,[91]) were also removed from all analyses. Next, data inclusion criteria were used to maximize the included dataset sizes and result generalizability, while also ensuring no considerable outlier (i.e., 4 standard deviations and more extreme than 99.9% of the distribution) biased results. Within these procedures, individual executive function measures were first screened for potential univariate leverage points in the association between age and each specific measure within general additive models (GAM: see below) or general additive mixed models (GAMM: see below). Leverage points were defined as those observations (measure for participant at testing session) with a residual from this model that was four standard deviations above the mean and removed from all subsequent analyses. Second, data were examined for potential multivariate outliers among all included executive function measures within each dataset using Mahalanobis distance within the *psych* package in R[92]. Sessions (all executive function measures for participant at testing session [i.e., study visit]) with a Mahalanobis distance four standard deviations above the mean were removed from all subsequent analyses.

## Data analysis
**General additive models.** General additive models (cross-sectional data: PNC dataset) and general additive mixed models (longitudinal data: Luna, NCANDA, NKI datasets) with penalized smooth plate regression splines via the *mgcv* package[41] were used to quantify nonlinear associations between age and executive function measures. Primary cross-sectional analyses (NKI, PNC) utilized a simple bivariate model examining the smoothed association between age (the independent variable) and executive function (the outcome measure). Primary longitudinal analyses (Luna, NCANDA) additionally included a smoothed term for visit number to account for potential non-developmental visit effects (e.g., practice: see Supplementary S9) and per-participant random intercepts and age slopes via *mgcv* GAMM. MGCV defaults were used for all parametrization with the exception that the maximum basis dimension for visit number in the NCANDA dataset was adjusted from 10 (the default) to 5 (given there were maximally five visits in this analysis dataset). Age-related fits from these primary GAM/GAMM models are presented in Fig. 1. Pointwise confidence intervals (displayed in Fig. 5) were generated by multiplying standard error estimates from the *mgcv* GAM/GAMM predict function by 2 and summing this with the predicted fit estimate. Sensitivity analyses (Supplementary S13–S17) examining sociodemographic and cognitive covariates followed the same procedures, with continuous variables (e.g., parental education) modeled as

smooth terms and categorical variables (e.g., biological sex) modeled as parametric terms.

**Periods of growth and maturational timing.** As in previous developmental research in different domains[49,50,87], periods of significant age-related change (age ranges) were defined by estimating the first derivative (finite differences method) in $1/10^{th}$ of a year intervals of GAM fits and performing a posterior simulation based on the GAM/GAMM model coefficients. Simultaneous (used given the multiple testing) confidence intervals (CI) were generated with the *gratia* package[51] with 10,000 simulations. Age ranges in which the simultaneous 95% CI did not include zero ($p < 0.05$) were classified as significant. Using this method, raster plots in Fig. 2 display color (red or blue) when the derivative is significant and white when the derivative is not significant. An aggregate analysis, pointwise three-level meta-analysis, incorporating all measures from all datasets was performed using the *metafor* package[93] with effects nested in measure and study. A cross-dataset label was used to nest measures from the same tasks (e.g., Penn CNB) across datasets. As in prior methodological work on point-wise meta-analysis with GAMs[94], meta-analytic estimates were computed across a common span of the independent variable: here, 1/10th year age bins, following linear interpolation of GAM/GAMM first derivatives. The same pointwise, three-level meta-analytic approach was used in aggregate analysis of GAM/GAMM fits in Fig. 5 and Supplementary Fig. S5. Secondary analyses that used an effect size threshold to define maturation scaled the GAM/GAMM fits from 0 (min) to 1 (max) to determine the percentage of total age-related that had occurred for each age (see Supplementary Fig. S5).

**Interdependence of performance across executive function tasks.** Cross-sectional and longitudinal correlations (linear, bivariate) were computed among executive function measures in each dataset (Fig. 3A). For longitudinal datasets (Luna, NCANDA), baseline refers to the first visit, longitudinal refers to the pooled within-person correlation via disaggregation with the *statsBy* function in the *psych* package in R. This approach was chosen to balance interpretability with model complexity for the accelerated longitudinal designs of Luna and NCANDA datasets. Aggregate analysis ("all measures") in Fig. 3A utilized a three-level meta-analysis via *metafor* with correlation pairs nested in task pairs and datasets. Exploratory factor analysis (Fig. 3B) via maximum likelihood method and a bifactor rotation was performed with the *psych* package in R from between- (Luna and NCANDA baseline and NKI, PNC datasets) and within-person correlation matrices (Luna and NCANDA longitudinal). Multiple data-driven thresholds for the number of extracted factors (Fig. 3C) were examined via parallel analysis and the *nScree* function in the *nFactors* R package[95] (95% CI from parallel analysis, factor analytic Kaiser rule, optimal coordinate, acceleration factor).

Contributions from domain-general versus specific processes to age-related differences in executive function were determined via model comparison that is also presented with the same description as well as additional visualization in Supplementary Fig. S8. To maximize comparability across studies and to prevent bias from shared, non-executive function visit effects (e.g., practice effects; see Sensitivity Analyses and Supplementary S9) analyses here were performed with cross-sectional data, although results are consistent with longitudinal data (cf., within-person factor structure in Fig. 3B, D).

First, three GAM models were fit for each dataset for each measure assessing the relationship between age and the specific measure i from subdomain x (measure$_{x\_i}$): model A, a composite metric created from all measures not in the same putative subdomain as measure$_{x\_i}$: composite metric $_{M \neq x}$, where $_{M \neq x}$ represents the set (M) of executive function measures that do not contain measures from subdomain x: model B, and a model where age is estimated from both measure$_{x\_i}$ and composite metric $_{M \neq x}$: model C.

As in primary analyses, the relationship between age and each measure was modeled with penalized splines. For each model (A–C), the percent of deviance explained in age was extracted (following standard estimation in *mgcv* GAM model). Next, the incremental deviance of age explained by measure$_{x\_i}$ over composite metric $_{M \neq x}$ was computed. Finally, the resulting measure specific age-related deviance was scaled to the original deviance estimate for the specific measure (model A) to create a percent of the original measure's age effect. The remaining percentage of model A's deviance was assigned as the domain-general percentage. To ensure consistent interpretability of the directionality of composite metric $_{M \neq x}$, measures from the opposing response type were sign flipped (e.g., latency sign flipped before creating equally weighted composite with accuracy measures). Sensitivity analysis examined the influence of the composite measure's precision in the estimation of domain-general accounts of age-related differences in executive function (see Supplementary Fig. S10).

**Normative maturational templates of age-related differences in executive function.** We used basis function regression with cross-validation ("leave one dataset out") to determine whether normative maturational templates of executive function could improve developmental inferences in new datasets and measures. A diagram of this procedure is likewise presented in Supplementary Fig. S11. In each iteration of the procedure, three (out of four) datasets were used to generate canonical executive function trajectories for accuracy and latency measures (measures aggregated across datasets via a pointwise three-level meta-analysis of GAM/GAMM age fits). The resulting output was then smoothed (via a subsequent GAM model), interpolated to the ages of the test ("left out") dataset, and fit as a single age parameter to each accuracy and latency measure of the left out dataset and compared to typical age models (age+age$^2$, inverse age [1/age], linear age [age]) as well as an intercept only (no age) model. Potential age models were evaluated with multiple metrics of model fit and complexity via the *performance* package in R[96] (longitudinal models [Luna, NCANDA]: $R^2$, adjusted $R^2$, Intraclass Correlation Coefficient [ICC], Root Mean Square Error [RMSE], residual standard deviation [Sigma], Akaike's Information Criterion [AIC], Bayesian Information Criterion [BIC]); cross-sectional models [NKI, PNC]): $R^2$, adjusted $R^2$, RMSE, Sigma, AIC, BIC). An additional sensitivity analysis explored the influence of the exact developmental timing of the developmental function with a similar procedure that offset in years (earlier or later) the canonical executive function trajectory (see Supplementary Fig. S12).

**Reporting summary**

Further information on research design is available in the Nature Portfolio Reporting Summary linked to this article.

## Data availability

This project used publicly available data for all analyses. Deidentified data for all datasets used in this project are available in public repositories pending appropriate data use agreements. Luna sample: nda.nih.gov/edit_collection.html?id=2831. NCANDA: ncanda.org (Release 4Y V02). NKI: fcon_1000.projects.nitrc.org/indi/enhanced/. PNC: ncbi.nlm.nih.gov/projects/gap/cgi-bin/study.cgi?study_id=phs000607.v3.p2. The data supporting the individual figures are provided in the Source Data Files. Summary data for the canonical executive function trajectory have been made available at https://github.com/tervoclemmensb/Executive_Function_Charting. Source data are provided with this paper.

## Code availability

Analysis code for the current project is available at https://github.com/tervoclemmensb/Executive_Function_Charting. Tervo-Clemmens, B., A Canonical Trajectory of Executive Function Maturation from

Adolescence to Adulthood, Executive Function Charting, https://doi.org/10.5281/zenodo.8302417, 2023.

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

## Acknowledgements

This work was supported by the National Institutes of Health: R03MH113090 (Calabro, Luna), R01MH067924 (Luna), an American Psychological Foundation Visionary Grant (Tervo-Clemmens), and the Staunton Farm Foundation (Luna).

## Author contributions

Conception: B.T.-C., B.L. Design: B.T.-C., F.J.C., A.C.P., B.L. Data acquisition, analysis, and interpretation: B.T.-C., F.J.C., A.C.P., J.F., W.F., B.L. Manuscript writing, revising: B.T.-C., F.J.C., A.C.P., J.F., W.F., B.L.

## Competing interests

The authors declare no competing interests.
