## [Peer Review File · Nature Communications]

A Canonical Trajectory of Executive Function Maturation from Adolescence to AdulthoodREVIEWER COMMENTS

Reviewer #1 (Remarks to the Author):

This was an exceptionally well-written report (especially in light of the complexity and density of findings) that utilizes four large developmental datasets to provide a reasonably definitive cognitive/EF maturation curve from adolescents until middle-young adulthood. The data analysis appears solid and reasonable and well-defined, and includes numerous sensitivity analyses. Thus, at each turn, when I would think "well what about language skill confounds" or "what about SES confounds in different catchments of these different studies" or "what about encroachment of mental illness in the dataset "(e.g. effects on reaction time), these concerns were proactively parried by numerous sensitivity analyses.

This work has clear take-aways, such as the fairly definitive inflection point of cognitive/EF development between 18-20 that IMO largely refutes the canon that EF is not developed until age 25 (just because of asymptotic timing of white matter volume trajectories). That cognition is for the most part maximal and sufficient by age 20 or so would probably seem more sensible to the problem-solvers storming and moving beyond Omaha Beach.

Second, the data are supportive of the Miyake/Friedman unity and diversity model that indicates a global EF factor that accounts for most shared variance in neurocognitive task prowess. I cannot comment on the appropriateness of the complex statistical models, but they come across as reasonably chosen and implemented.

My only suggestion (and it is not really a concern) would be that this rigorous amalgamation of both cross-sectional and longitudinal datasets has the additional potential to speak (statistically) to the utility of longitudinal designs for developmental studies, and how much more precision to at least these kinds of curve models they really add, despite their massive expense and complexity. Could the authors speak to that in a supplemental analysis? If not, this would be a ripe topic for a follow-on-paper.

Reviewer #2 (Remarks to the Author):

The authors present a study of the developmental trajectory of general executive function in adolescence using 4 large datasets – 2 longitudinal and 2 cross-sectional. I found the manuscript very well written, and believe it will be of potential high interest to many in public policy as well as cognitive development scientists. I do have a few comments and suggestions regarding clarity/novelty.

1) The authors neglect a fair amount of work on EF in childhood/adolescence in the behavioral genetics field (Clancy Blair, Elliot Tucker-Drob, etc.) and in education (Paul Cirino and others) that have large samples related to factoring EF into general and/or subcomponents. The authors note this work in adulthood, but work has been done in youth/childhood, and the manuscript would benefit from including this literature, even if the authors do not find much evidence for subcomponents of EF in their analysis. Example: Page 16 of combined PDF, line 576 section: "domain general EF development" discussion of unity/diversity.

2) I would appreciate a little clarification of how their "canonical EF" finding actually mathematically differs from an "age+age²" model comparison – the quadratic shape looks highly similar, and indeed appears to not be significantly different in fit estimate, especially for response times (Figure 5C-D). It seems like the authors have essentially used a novel approach to end up with a highly similar result for a fit that has been previously found by many others for EF abilities?

3) In general, I worry the authors are a bit dismissive of multiple societal influences on EF (SES, etc.) by looking only at a very zoomed out level of detail (single SES measure, general EF base models) – and that the media interpretation of this work will dismiss the large body of literature establishing how economic disparities do appear to have a significant impact on cognitive measures, and variability between individuals (e.g., Noble et al., 2015; Engelhardt et al., 2018).

4) The ages from 15-18 years are ambiguously discussed in the article – the authors emphasize growth from 10-15, but then skip in much of their discussion to 18-20. What is happening exactly in 15-18 – this seems critical to clarify for public policy regarding late-stage adolescents and cognitive maturity. If EF tasks continue to show growth through age 18-20, then why focus on 10-15? Basically, I would appreciate a bit more attention and clarity paid to ages 15-18/20, which is less studied in EF as a whole, and a big asset of this research manuscript.

5) The role of puberty on EF is not discussed, and these datasets seem perfect for having the power to explore this potentially critical factor of cognitive maturation in adolescence. It would be very interesting and important to combine pubertal status with the supplemental analysis of gender, as this is a known timing difference between genders. If there is no puberty data, this should be noted as a significant limitation.

6) I was also a bit confused as to why the supplemental analysis on sensitivity to socioeconomic factors only used parental education, when the authors also appear to have household income for the datasets (except for PNC). It would be helpful for interpretation if Supplemental Table S1 included parental education as well.

7) I would appreciate the methods including the gap between visits in the longitudinal datasets (average and range).

Figure 2: Related to the lack of clarity around ages 15-18, Figure 2 is hard to track specifics across tasks when the only age bar is all the way at the bottom of the figure. Is there a way to help the reader see the age for each task when growth is no longer significant?

Figure S4: It is very hard to tell the corrected line from all the stimuli lines – can something be done (i.e., put lines in color)?

Reviewer #3 (Remarks to the Author):

Thank you for the opportunity to review “A canonical trajectory of executive function maturation from adolescence to adulthood” submitted to Nature Communications. In this paper, the authors adopt a novel approach of integrating four independent North American datasets of age-graded/longitudinal performance on executive function tasks to estimate trajectories of normative development from adolescence to adulthood. Using non-linear modelling techniques, findings clearly demonstrate that executive function continues to develop into adolescence. Further, findings suggest that the development of executive function can be characterised by a domain-general executive function process.

The strengths of this study include the integration of datasets to create a large sample size and the inclusion of a broad range of executive function tasks. Previous research in this field has predominately been characterised by small sample sizes and/or a restricted range of executive function tasks. In my opinion, the most significant contribution of this study is the establishment of a foundation to model normative executive functioning during a period theorised to be critical to its development. This will provide a benchmark for future studies, including those examining

abnormalities in executive function development.

This paper is extremely well-written, and the attention to detail is impressive. In particular, the authors do a highly commendable job in describing and reporting the findings of sophisticated statistical techniques in a digestible format. The methodology is appropriate and above standard for the field; and is described in adequate detail for replication. In my opinion, this paper is worthy of publication in Nature Communications, and will likely become a touchstone for researchers in developmental neuropsychology broadly and especially for those focusing on executive function.

I only have four issues that might require further consideration prior to publication:

1. In the first paragraph of the introduction the authors note that protracted maturation and/or stabilisation of EF underlies lifespan peaks in risk-taking and increased vulnerability to psychiatric disorders. The reality is more complicated than this (i.e., not just dependent on the development of cognitive control), and the authors accurately provide references to studies suggesting that other factors are involved in the emergence of risk-taking and psychiatric vulnerability. Rather than just providing references, I feel that an additional sentence is warranted to acknowledge that cognitive control is only one domain of development/functioning that is involved in risk-taking (e.g., sensation seeking) or psychiatric vulnerability.
2. The authors use a "data-driven" approach to characterise trajectories of executive function development through the use of general additive models. A similar data-driven approach is taken to identify latent factors in executive function task performance using factor analysis. Almost all factor analytic criteria identify more than one latent factor in the analysis described here, but the authors select a single latent domain-general factor to characterise performance. They justify this by correctly identifying that a single factor accounts for the majority of variance in task performance. However, this is always the case for factor analytic methods – the first factor will almost always account for the majority of variance. This is a widely debated area in the research field of executive functioning (unity vs. diversity), and likely to be a point of contention for researchers in the field reading this paper. Given this, it might be useful to provide more detail/justification about why a single factor was chosen, and/or provide some additional supplementary details about models that had more than one factor (e.g., how did such models perform in characterising age-graded changes?)
3. Ethnicity was not explored as a potentially moderating variable of the findings in the sensitivity analyses – is there a justification for not doing this?
4. As a limitation of the study, it would be appropriate to highlight that the data and findings only relate to North American samples and may not necessarily generalise to other locations or individuals from developing Nations.

My comments are minimal, which reflects the quality of this work - the authors are to be commended for an excellent and thorough piece of work!

Best,
James Ogilvie

We thank the editor and the reviewers for the helpful feedback and the opportunity to revise our manuscript. We have strengthened the manuscript overall and provide detail on our changes below each of the reviewer's comments.

REVIEWER COMMENTS

Reviewer #1 (Remarks to the Author):

This was an exceptionally well-written report (especially in light of the complexity and density of findings) that utilizes four large developmental datasets to provide a reasonably definitive cognitive/EF maturation curve from adolescents until middle-young adulthood. The data analysis appears solid and reasonable and well-defined, and includes numerous sensitivity analyses. Thus, at each turn, when I would think "well what about language skill confounds" or "what about SES confounds in different catchments of these different studies" or "what about encroachment of mental illness in the dataset "(e.g. effects on reaction time), these concerns were proactively parried by numerous sensitivity analyses.

We thank the reviewer for their very positive feedback regarding the manuscript, the writing, and our goal of providing a comprehensive and rigorous set of analyses towards a novel, precise quantitative charting, multi-assessment, and multi-dataset investigation and replication of executive function development from adolescence to adulthood.

This work has clear take-aways, such as the fairly definitive inflection point of cognitive/EF development between 18-20 that IMO largely refutes the canon that EF is not developed until age 25 (just because of asymptotic timing of white matter volume trajectories). That cognition is for the most part maximal and sufficient by age 20 or so would probably seem more sensible to the problem-solvers storming and moving beyond Omaha Beach.

We thank the reviewer for highlighting the impact of our work here and agree, this was a motivating factor for our investigation of using multiple large scale datasets to move towards a comprehensive and consensus position on this key aspect of development with critical implications.

Second, the data are supportive of the Miyake/Friedman unity and diversity model that indicates a global EF factor that accounts for most shared variance in neurocognitive task prowess. I cannot comment on the appropriateness of the complex statistical models, but they come across as reasonably chosen and implemented.

We thank the reviewer for recognizing the importance of this and testing such a prominent theory at scale, that had yet to be tested in development.

My only suggestion (and it is not really a concern) would be that this rigorous amalgamation of both cross-sectional and longitudinal datasets has the additional potential to speak (statistically) to the utility of longitudinal designs for developmental studies, and how much more precision to at

least these kinds of curve models they really add, despite their massive expense and complexity. Could the authors speak to that in a supplemental analysis? If not, this would be a ripe topic for a follow-on-paper.

This is an important point. In the revised manuscript we now expand upon the value of such a potential follow-up investigation that examines the relative utility of the longitudinal versus cross-sectional data to this specific question, which includes consideration of research design, data acquisition costs, statistical power, and potential non-developmental visit effects.

“Future work may also use these insights towards optimizing developmental study designs with respect to the number of participants, construct breadth of assessments, and the number of longitudinal time points.”

Reviewer #2 (Remarks to the Author):

The authors present a study of the developmental trajectory of general executive function in adolescence using 4 large datasets – 2 longitudinal and 2 cross-sectional. I found the manuscript very well written, and believe it will be of potential high interest to many in public policy as well as cognitive development scientists. I do have a few comments and suggestions regarding clarity/novelty.

We thank the reviewer for the overall positive impression and helpful comments that we have clarified in the revised manuscript .

1) The authors neglect a fair amount of work on EF in childhood/adolescence in the behavioral genetics field (Clancy Blair, Elliot Tucker-Drob, etc.) and in education (Paul Cirino and others) that have large samples related to factoring EF into general and/or subcomponents. The authors note this work in adulthood, but work has been done in youth/childhood, and the manuscript would benefit from including this literature, even if the authors do not find much evidence for subcomponents of EF in their analysis. Example: Page 16 of combined PDF, line 576 section: “domain general EF development” discussion of unity/diversity.

We thank the reviewer for pointing us to additional relevant citations and we have practiced due care to incorporate both work pointed out by the reviewer and related work from this area and in ways that such references may be pointed to in additional clarification points (cf., Reviewer 2 Point 3).

While prior work in adults^{50,64} and younger children⁶⁵⁻⁶⁷ has provided evidence for a potential unity/diversity framework of executive function, the relative domain-general versus specificity of executive function has not been examined with respect to changes during adolescence.

50. Tucker-Drob, E. M. Global and domain-specific changes in cognition throughout adulthood. *Developmental psychology* **47**, 331 (2011).

- 64 Miyake, A. & Friedman, N. P. The nature and organization of individual differences in executive functions: Four general conclusions. *Current directions in psychological science* **21**, 8–14 (2012).
65. Cirino, P. T. *et al.* A framework for executive function in the late elementary years. *Neuropsychology* **32**, 176 (2018).
66. Blair, C., Zelazo, P. D. & Greenberg, M. T. The measurement of executive function in early childhood. *Developmental neuropsychology* **28**, 561–571 (2005).
67. Blair, C. Developmental science and executive function. *Current directions in psychological science* **25**, 3–7 (2016).

2) I would appreciate a little clarification of how their “canonical EF” finding actually mathematically differs from an “age+age²” model comparison – the quadratic shape looks highly similar, and indeed appears to not be significantly different in fit estimate, especially for response times (Figure 5C-D). It seems like the authors have essentially used a novel approach to end up with a highly similar result for a fit that has been previously found by many others for EF abilities?

- A. This is an important point to clarify. First, a primary goal of this work (intended for the special collection on reproducibility and generalizability) is indeed to provide a comprehensive “multi-assessment, and multi-dataset investigation and *replication* of EF development from adolescence to adulthood”. Thus, the novelty here is both in the scope and scale of the project towards reproducibility, generalizability, and real-world utility as well as approaches to address prior challenges in directly quantifying maturational timing. We now better clarify this in the revised introduction:

“Here we aggregated four large-scale, independent datasets to construct the most comprehensive set of executive function data spanning the entire adolescent period as well as the relative transitional periods of late childhood and early adulthood (total age range: 8-35, total N=10,766, total visits =13,817) that included 23 executive function measures from 17 distinct tasks. In addition to large-scale replication, we directly address prior challenges in defining the maturational timing and domain-generality of adolescent executive function development with multiple large independent cohorts (two longitudinal, two cross-sectional), nonlinear modeling approaches that directly define maturational timing, and the inclusion of a comprehensive executive function battery that permitted the investigation of both potential unitary and diversity processes.”

- B. As we now better clarify, our primary analytic approach for estimating developmental trajectories used penalized splines via semiparametric general additive models/ general additive mixed models to arrive at a data-driven functional form of age. This means that the observed functional form of the trajectory is part of the modeling process and definitionally the best fit that balances complexity and prevents overfitting via penalization. This has several advantages over traditional parametric approaches (linear, inverse linear [$1/\text{age}$], or quadratic [$\text{age} + \text{age}^2$]). Unlike traditional fixed-parametric approaches, inferences can be drawn on the shape of the trajectory and therefore the maturational timing. Most notably, comparing common functional forms of age, the data-driven and recovered trajectory has a clear asymptote, characterization of which is not possible with a quadratic functional form of age. See for example Figure R1 below where

standard functional forms are compared to a composite metric of executive function across datasets.

Figure R1. *Comparison of parametric functional forms of age to data-driven GAM.*

- By definition, the parameterization from a quadratic model for executive function that increases with a rapid development followed by stabilization towards adult levels will force the trajectory to reverse course and start decreasing (see Figure R1 above). The inaccuracies and challenges in interpretability of standard, parametric functional forms are highlighted further by examining areas of the local change (first derivative). Whereas by definition the quadratic model doesn't allow for an asymptote and actually will as age increases demonstrate age-related decreases, the data-driven GAM model recovers such an asymptotic feature.

Figure R2. *Comparison of derivatives from parametric functional forms of age to data-driven GAM. Grey line indicates zero.*

- The asymptotic feature of the data-driven recovery of a canonical trajectory of executive function development is essential for interpretability and accurate inferences across the lifespan, but also for uniquely allowing for a “*systematic investigation of the relative rate and timing of maturation that is essential for precise developmental science*”. For example, using the properties of this non-linear but asymptotic data-driven trajectory we are able to precisely quantify with simulation age periods of significant local age-related changes (via the first derivative) (Figure 2; pasted below as Figure R4 for ease). Consistent with the special collection on reproducibility and generalizability in *Nature Communications*, and as Reviewer 1 highlights, a primary goal was to test the reproducibility and generalizability of this key notion of precise developmental trajectory and maturational timing of executive function.

Figure R3 (same as Figure 2 in primary manuscript, reproduced for reviewer ease; *see primary manuscript for full figure legend*). **Developmental Periods with Significant Age-related Change in Executive Function.** Raster plots display color if executive function measure has significant ($p < .05$ [two-sided] via simultaneous confidence intervals from posterior simulation of first derivative of GAM/GAMM fits; *see Methods*) age-related change at that age (red: significant age-related increases; blue: significant age-related decreases; color scale is dataset standard deviation units)

C. We now better clarify that Figure 5 is the one place the reviewer does mention in the primary manuscript where we do statistically compare the recovered canonical trajectory to established parametric models of age. We clarify however, that this is not a typical model comparison, as if it were, by definition the data-driven trajectory would always perform best, as it is penalized to be the best fit to the data (cf., Extended Data Figure 2 for individual best fits and meta-analytic trajectories). Rather, we use a simplified version of the data-driven trajectory (a single parameter basis function) developed in the independent datasets and rigorous *out-of-dataset estimation* as a test not of the accuracy of the trajectory itself for the data it was generated from (which is guaranteed based on the fitting and meta-analytic procedures) but its ability, even in a simplified form to improve inferences in NEW datasets and NEW measures (through our distributed summary data: see Data Availability which links to this summary data that can be used directly in future work:

https://github.com/tervoclemmensb/Executive_Function_Charting/blob/main/Figures/Metabasis/Metabasis.acclat.data.csv). That is, how well does this shape (developed in independent samples with different measures) predict the shape of a new executive function measure.

- Such use of independently derived empirical basis functions has a long history in signal processing and for example neuroimaging analyses but has not been done and could not have been done (given the lack of multi-dataset multi-assessment studies of executive function in this area) before in this literature.
- Based on the reviewer comments we have revised this section to be more clear, however.
 - In this specific set of analyses that test the potential for the results here to directly quantitatively guide future research (via our distributed code and summary data), this rigorous out of sample work shows that as stated in our manuscript:

“Following model selection criteria based on all metrics across accuracy and latency measures, the simplified, single parameter basis function was the most selected model (55.6% of the time; compare to quadratic [$age + age^2$]: 37.3%; inverse linear age [$1/age$]: 7.03%; linear age: 0%), which was significantly higher than all other age models (vs. inverse linear age [$1/age$]: $\chi^2=21.6$, $p < .001$; vs. linear age $\chi^2=30.6$, $p < .001$; all p -values two sided) other than the quadratic model (vs. $age + age^2$ (best model 37.3%), $\chi^2=2.29$, p [two-sided] = .130).”

- Therefore it is not that the quadratic model was better or that it was similar (see again Figures R1, R2), our rigorous validation just showed that we could not *statistically* distinguish between a highly simplified version of our model (1 parameter) and a quadratic (2 parameters), in this very specific most “severe test” (cf., Yarkoni, T. (2022). The generalizability crisis. *Behavioral and Brain Sciences*, 45, e1.) of, again not the fit itself, but the generalizability of this simplified single parameter canonical trajectory to apply to any measure from any dataset (again via available distributed summary data:

https://github.com/tervoclemmensb/Executive_Function_Charting/blob/main/Figures/Metabasis/Metabasis.acclat.data.csv).

- To better clarify this point we have revised this section of the manuscript:

“Combined, these results establish a simplified, single parameter data-driven basis function version of the canonical executive function trajectory as an alternative, developmentally informed functional form of age that is superior or highly competitive with standard, parametric functional forms of age when applied to new datasets and new measures. Therefore, we suggest that, along with full, multi-parameter complex spline models (GAM/GAMMs used throughout the rest of the manuscript) and standard functional forms of age (e.g., linear, inverse linear, quadratic), such a simplified, developmentally informed basis function may quantitatively (see Data and Code Availability) inform future research on normative development and deviations from normative development in health and disease (see Discussion).

- We also now better clarify however when discussing the ability of future work to be directly quantitatively guided by this research by a simple basis function (in addition to the widespread replication, key conceptual insights of demarcating the adolescent period) that further model testing should likewise be performed in an individual dataset.

“To assist in this pursuit, we have made available summary data (note participant-level data is also available with necessary data use agreements; see Data availability) for the canonical executive function trajectory, with the goal that subsequent work may utilize and continue to refine empirically defined normative maturational templates in executive function research. While part of this refinement should include model comparison of other candidate functional forms of age (cf., Figure 5), we suggest sharing of reproducible and well-powered adolescent trajectories of executive function can be directly integrated in future analysis (e.g., basis function regression) in methods that mirror the development, refinement, and use of summary statistics in other fields (e.g., polygenic risk scores⁶⁰).”

3) In general, I worry the authors are a bit dismissive of multiple societal influences on EF (SES, etc.) by looking only at a very zoomed out level of detail (single SES measure, general EF base models) – and that the media interpretation of this work will dismiss the large body of literature establishing how economic disparities do appear to have a significant impact on cognitive measures, and variability between individuals (e.g., Noble et al., 2015; Engelhardt et al., 2018).

- We appreciate and share the reviewer’s concern as to avoid this misinterpretation of our work. We have modified the text to make this clearer. We in fact make no claims about the potential direct impact of SES on EF, rather we are demonstrating that our results are not biased by unmodeled potential SES differences within individual samples. As

detailed in our sensitivity analyses, we show that when adjusting for SES indicators, the shape and timing of the canonical EF trajectory does not change. This is a demonstration that our observed results are not confounded by SES differences, for example. We do not, however, compare SES strata on EF, as this is outside of the scope of this work, although we do believe this work is important for the field and have incorporated these suggestions into the manuscript.

- In the revised manuscript we now better clarify this explicitly, and mention how the insights and quantitative tools generated here could be used to study SES differences (something we did not do) as well:

“Sensitivity analyses demonstrated that primary results concerning the magnitude and timing of executive function accuracy and latency development were consistent across males and females (Supplemental S6). Additional sensitivity analyses demonstrated that our primary results did not change when covarying for socioeconomic indicators (parental education and family income, Supplemental S7,S8) and assessments of culturally acquired knowledge (verbal reasoning and vocabulary, Supplemental S9), and remained consistent across mental health inclusion/exclusion thresholds (Supplemental S10). This suggests that mathematically holding these factors constant did not change the current results that focused on aggregate and average executive function changes during adolescence. Thus, our results do not speak to for example past findings suggesting economic disparities impact cognitive measures, and variability between individuals (cf.,^{55,56}). However, the tools and insights from the current work can be used for future studies focused on relationships between these factors and executive function in more detail (see Discussion). As in previous longitudinal investigations of computerized and neuropsychological performance⁵⁷, age-independent visit effects (e.g. practice effects) on cognitive testing were observed for many executive function tasks in longitudinal samples (Supplemental S5). However, all longitudinal analyses (Luna, NCANDA samples) covaried for a non-linear effect of visit number (see Methods) and we demonstrate replication to two cross-sectional datasets (NKI, PNC) where visit effects could not have occurred, indicating that our primary results are likewise robust to practice effects on cognitive testing.”

“Such domain-general executive function development may help explain, for example, wide-spread differences across executive function tasks in clinical^{68,69} and/or population research (e.g., social determinants of health^{67,70,71}), as well as the tendency for many executive function tasks to engage common neural circuitry^{72,73}.”

4) The ages from 15-18 years are ambiguously discussed in the article – the authors emphasize growth from 10-15, but then skip in much of their discussion to 18-20. What is happening exactly in 15-18 – this seems critical to clarify for public policy regarding late-stage adolescents and cognitive maturity. If EF tasks continue to show growth through age 18-20, then why focus on 10-15? Basically, I would appreciate a bit more attention and clarity paid to ages 15-18/20, which is less studied in EF as a whole, and a big asset of this research manuscript.

- We thank the reviewer for this important point and indeed 15-18 years old represents the period of the “elbow” in the canonical non-linear trajectory and thus represents a key period of development to consider.
- This represents the period of development through which small but still significant EF changes can be observed.
- We now better clarify this developmental period of mid adolescence within these nonlinear trajectories (again uniquely afforded to the nonlinear modeling here [see point 1]): several times in the revised manuscript:

“These analyses revealed that nearly all executive function measures (20/23 measures) had corrected significant (corrected p 's < .004, [two-sided]; see Supplemental S2 for full statistics as well reproducible variable names from public datasets) age-related differences and followed a highly similar non-linear developmental trajectory, with rapid development in late childhood to mid-adolescence (10-15-years-old), smaller changes through mid-adolescence (15-18-years-old), before stabilizing to adult-levels in late adolescence (18-20-years-old) (Figure 1A-D).”

“From middle to late adolescent periods (15-18-years-old), smaller but still statistically significant changes were observed for several measures (Figure 2A-D).”

“This suggests that while adolescents clearly possess complex cognitive abilities, including the ability to inhibit prepotent responses, maintain and update information in memory, and abstractly plan for future events, such abilities do not reach their full potential until 18-20-years-old (late adolescence). Adolescent periods prior to this age-range (i.e., early to mid-adolescence ~10-15-years-old, and mid to late adolescence ~15-18-years old) are therefore likely critical final stages of this type of cognitive development, where deviations from normative development may lead to poorer outcomes in adulthood.”

5) The role of puberty on EF is not discussed, and these datasets seem perfect for having the power to explore this potentially critical factor of cognitive maturation in adolescence. It would be very interesting and important to combine pubertal status with the supplemental analysis of gender, as this is a known timing difference between genders. If there is no puberty data, this should be noted as a significant limitation.

- We agree this is an important point. Disambiguating pubertal development from age-related changes (where age-related change was the primary goal of these analyses and key to the upper bound of the adolescent period) is very challenging, particularly in the context of large, cross-sectional age differences. We agree with the reviewer and have added this as a limitation of the current results and point to this as an important area for follow up work.

“Our work also sets key areas for future work regarding maturational timing in more fundamental studies of executive function development (e.g., disambiguating age-related change from pubertal development, generalizability to populations outside of the United States, targets for brain imaging, and considerations for affective versus non affective executive function tasks: see Supplemental Discussion).”

“Another potential limitation is that the current work did not try to disambiguate age-related changes from pubertal development, given challenges in independently estimating these effects in the presence of large cross-sectional age effects (cf., ⁹²). However, it will be important for future work, particularly when focusing on early periods of adolescence to likewise seek large-scale multi-assessment, multi-dataset reproducibility for the specific role of pubertal status in driving executive function development.”

6) I was also a bit confused as to why the supplemental analysis on sensitivity to socioeconomic factors only used parental education, when the authors also appear to have household income for the datasets (except for PNC). It would be helpful for interpretation if Supplemental Table S1 included parental education as well.

- Per the reviewer's suggestion we have added a sensitivity analysis covarying for household income, as Supplemental S7 (reproduced here as Figure R4 for reviewer ease).

Figure R4 (same as Supplemental S7 in primary manuscript, reproduced for reviewer ease; see primary manuscript for full figure legend). **The Magnitude and Timing of Executive Function Development is Consistent with Primary Results when Statistically Covarying for Parental Education.** Subtle differences in the magnitude of executive function accuracy, but not the timing, were observed when covarying for family income in NCANDA, compared to the base model. However, these differences did not replicate across samples, as the magnitude and timing of executive function development were nearly identical for Luna and NKI datasets when covarying for family income.

We have also added parental education to Table S1 (reproduced here for reviewer ease)

	Dataset	Luna	NCANDA	NKI	PNC	ACS 2011-2015
Characteristic		%	%	%	%	%
Race/Ethnicity	NH White	78.7	64.1	53.3	56.2	59.8
	NH Black/AA	12.2	11.9	19.0	35.6	11.7
	Asian, AIAN, NHPI	3.2	8.2	9.1	0.4	7.0
	Hispanic	5.9	11.6	16.9	5.8	12.3
	Other	—	4.3	1.7	2.0	9.3
Sex	Male	47.4	49.1	51.5	47.9	50.9
	Female	52.6	50.9	48.5	52.1	49.1
Family Income	<25k	0.0	6.0	16.0	—	24.6
	25k-49k	8.3	11.3	19.8	—	22.2

	50k-74k	13.5	12.9	18.6	—	17.1
	75k-99k	21.2	13.5	12.9	—	12.1
	100k-199k	35.3	34.5	24.8	—	18.8
	200k+	21.7	21.9	8.1	—	5.3
Guardian Highest Education	Incomplete High School	< 1	2.7	4.5	4.4	—
	High School	14.2	6.0	22.0	27.1	—
	1-3 Years of College	29.5	16.1	23.8	24.9	—
	Bachelor's	31.3	35.7	27.8	26.9	—
	Postgraduate	23.3	39.6	21.9	16.5	—

Note. Characteristic coding for race/ethnicity, sex, and family income based on recent, large-scale multi-site developmental approach⁹³. Non-Hispanic (NH). African American (AA). American Indian and Alaska Native (AIAN). Native Hawaiian and Pacific Islander (NHPI). Data unavailable/not collected (—). American Community Survey (ACS) displays population estimates of the United States. Guardian Highest Education based on explicit identification of given categories (Luna) or approximation based on years of reported education (incomplete High School < 12, High School = 12, 1-3 Years of College = 13-15, Bachelor's = 16, Postgraduate > 16). See Methods for information on recruitment. See Extended Data Figure 1 for age distributions of participants by study visit.

7) I would appreciate the methods including the gap between visits in the longitudinal datasets (average and range).

We have added this information to the manuscript. We likewise note that the primary analytic approach utilized participant age as a fully continuous metric of time and through the multilevel modeling approach via maximum likelihood estimation permits sparsity at the per-participant level. The visit structure by age can be visualized in our Extended Data Figure 1, reproduced here as Figure R5 for reviewer ease.

For Luna dataset: “...(total participant visits=666, median number of visits per-participant= 3; range of visits per-participant=1-10; median months between visits=13.3; range of months between visits=5.97-81.73; see Figure 1 for graphical depiction of dataset by visit structure)”.

For NCANDA dataset: “...(total participant visits= 3,412, median number of visits per-participant= 5; range of visits per-participant=1-5; median months between visits=12.17; range of months between visits=4.98-23.97; see Extended Data Figure 1 for graphical depiction of dataset age by visit structure)”

Figure R5 (same as Extended Data Figure 1 in primary manuscript, reproduced for reviewer ease; see primary manuscript for figure legend). *Age Range and Study Design of Datasets.*

Figure 2: Related to the lack of clarity around ages 15-18, Figure 2 is hard to track specifics across tasks when the only age bar is all the way at the bottom of the figure. Is there a way to help the reader see the age for each task when growth is no longer significant?

We clarify that the “all measures” bar in Main Text Figure 2E (Full Figure Displayed in this response as Figure R3, relevant section pasted below) provides a fully continuous weighted aggregation across measures and datasets (three-level meta-analysis) as a function of age.

E. All Measures

Per the reviewer's suggestion, we have added an additional visualization to the supplement to specifically visualize the end points across measures.

Figure R6 (same as Main Text Supplemental S3, reproduced here for reviewer ease). ***Final Age of Significant Age-Related Change (via First Derivative) for All Executive Function Measures.*** To assist in the visualization of Main Text Figure 2, histograms display the final age (in years) of detected statistical significance for the first derivative in all accuracy (left, red) and latency (right, blue) executive function measures. The hashed lines indicate the median across measures. See Main Text and Methods for details on significance testing, Main Text Figure 2E for fully continuous aggregate analysis (three-level meta-analysis) of significant periods of age-related change.

Figure S4: It is very hard to tell the corrected line from all the stimuli lines – can something be done (i.e., put lines in color)??

We now better clarify that the line type itself indicates whether the model for this non-developmental visit effect (e.g., practice) was significant. Consistent with prior work, only three models (see Figure R4, Luna: MIX accuracy, SOC latency; NCANDA: PNBK latency) did not show corrected significant non-developmental visit effects and we now better specify this in the Figure caption and have increased the size of the figure to likewise assist the reader.

A. Luna

B. NCANDA

Figure R4 (same as Figure S4 in primary manuscript, reproduced for reviewer ease). **Non-Developmental, Visit Effects are Evident in Longitudinal Executive Function Measures.**

All measures scaled to per-dataset standard deviation (z) units. A) Non-linear fits for visit effects from the Luna dataset (N=196; 668 total visits) of general additive mixed model (GAMM; multilevel penalized spline regression) for Antisaccade (ANTI), Fixation Breaks (FIX), Mixed Antisaccade (MIX), Spatial Span (SSP), Delayed Matching to Sample (DMS), Memory Guided Saccade (MGS), Stockings of Cambridge (SOC), and equally weighted composite metrics (z score sum of all accuracy, latency measures; COMP) for accuracy measures (top) and latency (bottom). All models covaried for a smoothed effect of age. Solid line indicates models with Bonferroni corrected significance. Dashed indicates models that do not surpass this threshold. B) Non-linear, GAMM fits for visit effects from the NCANDA dataset (N=831; 3,412 total visits) for Penn Conditional Exclusion (PCET), Penn Continuous Performance (PCTP), Penn N Back (PNBK), and Stroop (STRP) tests and equally weighted accuracy, latency composite metrics (COMP). All models covaried for a smoothed effect of age. Solid line indicates models with Bonferroni corrected significance. Dashed indicates models that do not surpass this threshold (Luna: MIX accuracy, SOC latency; NCANDA: PNBK latency). As in previous longitudinal investigations of computerized and neuropsychological performance, age-independent visit effects (e.g., practice effects) on cognitive testing were observed for many executive function tasks. However, given that all longitudinal analyses covaried for these non-linear, smoothed effects of visit and our primary analyses and inference were additionally based on two cross-sectional, independent replication datasets (NKI, PNC), the primary results from this manuscript are judged to be robust to non-developmental visit (e.g., practice) effects.

Reviewer #3 (Remarks to the Author):

Thank you for the opportunity to review “A canonical trajectory of executive function maturation from adolescence to adulthood” submitted to Nature Communications. In this paper, the authors adopt a novel approach of integrating four independent North American datasets of age-graded/longitudinal performance on executive function tasks to estimate trajectories of normative development from adolescence to adulthood. Using non-linear modelling techniques, findings clearly demonstrate that executive function continues to develop into adolescence. Further, findings suggest that the development of executive function can be characterised by a domain-general executive function process.

The strengths of this study include the integration of datasets to create a large sample size and the inclusion of a broad range of executive function tasks. Previous research in this field has predominately been characterised by small sample sizes and/or a restricted range of executive function tasks. In my opinion, the most significant contribution of this study is the establishment of a foundation to model normative executive functioning during a period theorised to be critical to its development. This will provide a benchmark for future studies, including those examining abnormalities in executive function development.

This paper is extremely well-written, and the attention to detail is impressive. In particular, the authors do a highly commendable job in describing and reporting the findings of sophisticated statistical techniques in a digestible format. The methodology is appropriate and above standard for the field; and is described in adequate detail for replication. In my opinion, this paper is worthy of publication in Nature Communications, and will likely become a touchstone for researchers in developmental neuropsychology broadly and especially for those focusing on executive function.

We thank the reviewer for the positive review of the manuscript and highlighting key areas of strength regarding the large-scale, multi-dataset investigation, the use of non-linear modeling approaches, and replication.

I only have four issues that might require further consideration prior to publication:

1. In the first paragraph of the introduction the authors note that protracted maturation and/or stabilisation of EF underlies lifespan peaks in risk-taking and increased vulnerability to psychiatric disorders. The reality is more complicated than this (i.e., not just dependent on the development of cognitive control), and the authors accurately provide references to studies suggesting that other factors are involved in the emergence of risk-taking and psychiatric vulnerability. Rather than just providing references, I feel that an additional sentence is warranted to acknowledge that cognitive control is only one domain of development/functioning that is involved in risk-taking (e.g., sensation seeking) or psychiatric vulnerability.

We thank the reviewer for raising this very important point and we absolutely agree. We have added this clarifying point in our first mention of such etiological considerations (First paragraph, third sentence):

In parallel to socioemotional development and environmental influences, a protracted maturation^{7,8} and/or stabilization of executive function⁹ into adulthood has been suggested to contribute to lifespan peaks in risk-taking behaviors (e.g., substance use initiation¹⁰; though see also^{11,12}) and increased vulnerability to psychiatric disorders¹³ during adolescence.

2. The authors use a “data-driven” approach to characterise trajectories of executive function development through the use of general additive models. A similar data-driven approach is taken to

identify latent factors in executive function task performance using factor analysis. Almost all factor analytic criteria identify more than one latent factor in the analysis described here, but the authors select a single latent domain-general factor to characterise performance. They justify this by correctly identifying that a single factor accounts for the majority of variance in task performance. However, this is always the case for factor analytic methods – the first factor will almost always account for the majority of variance. This is a widely debated area in the research field of executive functioning (unity vs. diversity), and likely to be a point of contention for researchers in the field reading this paper. Given this, it might be useful to provide more detail/justification about why a single factor was chosen, and/or provide some additional supplementary details about models that had more than one factor (e.g., how did such models perform in characterising age-graded changes?)

- This is an important point. In the revised manuscript, we now better clarify our justification for highlighting a single factor, while also making clear our exploration of additional factors.

For example, Main Text Figure 3B (pasted below for reviewer ease) demonstrates the amount of variance explained by each extracted factor for each dataset (Top) and the support for these factors based on a number of data-driven thresholds (parallel analysis, optimal coordinate, acceleration factor, and a factor analytic Kaiser rule; see Methods).

B. Factor Analysis: Variance Explained, N Factors Supported

- We also demonstrate the factor loadings for the First three factors for all datasets in Extended Data Figure 4 (Pasted below for reviewer ease).

- Finally, we provide correlation matrices for all variables for all datasets (Supplemental S4) to support further empirical and meta-analytic work considering factor structure, consistent with our aim of providing quantitative tools to guide future research in this area.
- In the revised manuscript, we also now make clear the value of additional work exploring the factor structure and that while our results most strongly support and are well accounted for by a domain general (unity) perspective of executive function future work should further explore this issue. In particular, we suggest that larger multi dataset work with harmonized measures will be useful towards these goals.

“Although we found a considerable degree of commonality in adolescent executive function development, as in related work from adults⁴², current measures and methods do not rule out additional executive function variance relevant to development (even if such domain/measure-specific variance is less prominent than domain-general processes). Our analyses were generally well accounted for by a domain general perspective of executive function, and further exploring this allowed us to examine multi-assessment multi-dataset meta-analytic estimates towards reproducibility and generalizability. However, as in other reports^{42,66–68}, executive function variance was not entirely captured by a single factor. Future work, including using the tools and insights developed here,

may be used to address these questions in multi-dataset reproducibility and generalizability investigations.”

3. Ethnicity was not explored as a potentially moderating variable of the findings in the sensitivity analyses – is there a justification for not doing this?

We now better clarify we examined sample composition (including race/ethnicity, sex, age, and family income) via inverse propensity weighting to match sociodemographic census data from the United States (as detailed in Supplemental 1; pasted below for reviewer ease). While we appreciate the potential value in understanding cultural differences and generalizability therein (see important limitation added based on feedback from the reviewer below), we believe isolated and decontextualized moderation by race/ethnicity has problems of interpretability (given stratification of certain racial/ethnic groups with respect to for example income in the United States) and a strong risk for misinterpretation and bias.

Figure S1. Executive Function Development is Consistent with Primary Results when Weighting to Match the American Community Survey

Plots display age trajectories of domain general accuracy and latency executive function measures (equally weighted composite metrics, z score sum of all accuracy, latency measures) for base GAM models for NKI and PNC datasets. Grey plots are from primary analyses used throughout the manuscript. Light blue plots are design-based analyses utilizing inverse propensity weighting to match sociodemographic characteristics of the 2011-2015 American Community Survey (see Supplemental Table S1). For NKI, the variables used in the weighting procedure were race/ethnicity, sex, age, and family income. For PNC, the variables used in weighting procedure were race/ethnicity, sex, and age.

4. As a limitation of the study, it would be appropriate to highlight that the data and findings only relate to North American samples and may not necessarily generalise to other locations or individuals from developing Nations.

We thank the reviewer for this suggestion and have added it to the manuscript:

“Our work also sets key areas for future work regarding maturational timing in more fundamental studies of executive function development (e.g., disambiguating age-related change from pubertal development, generalizability to populations outside of the United States, targets for brain imaging, and considerations for affective versus non affective executive function tasks: see Supplemental Discussion).”

“Relatedly, while the aggregated datasets and inferences drawn here appear to approximate population patterns from the United States, further work with multinational and multicultural samples is required to determine the generalizability of these results to other countries and cultures.”

My comments are minimal, which reflects the quality of this work - the authors are to be commended for an excellent and thorough piece of work!

We thank the reviewer for the positive impression and share their excitement for the potential impact of this work.

REVIEWERS' COMMENTS

Reviewer #1 (Remarks to the Author):

The authors have done a commendable job in addressing reviewer concerns, especially regarding limitations of the dataset and future directions. This will be a welcome addition to the developmental literature.

Reviewer #2 (Remarks to the Author):

The authors were very responsive to my comments and I find the edits made very helpful in reducing confusion/misinterpretation. I commend the authors on this tremendous work.

Reviewer #3 (Remarks to the Author):

The authors have done a thorough job of addressing the reviewer comments and am satisfied with their changes/responses. I have no further comments to add.